# Control of neuronal excitation–inhibition balance by BMP–SMAD1 signalling

Zeynep Okur[1], Nadia Schlauri[1,4], Vassilis Bitsikas[1], Myrto Panopoulou[1], Raul Ortiz[1], Michaela Schwaiger[2,3], Kajari Karmakar[1,5], Dietmar Schreiner[1] & Peter Scheiffele[1✉]

Throughout life, neuronal networks in the mammalian neocortex maintain a balance of excitation and inhibition, which is essential for neuronal computation[1,2]. Deviations from a balanced state have been linked to neurodevelopmental disorders, and severe disruptions result in epilepsy[3–5]. To maintain balance, neuronal microcircuits composed of excitatory and inhibitory neurons sense alterations in neural activity and adjust neuronal connectivity and function. Here we identify a signalling pathway in the adult mouse neocortex that is activated in response to increased neuronal network activity. Overactivation of excitatory neurons is signalled to the network through an increase in the levels of BMP2, a growth factor that is well known for its role as a morphogen in embryonic development. BMP2 acts on parvalbumin-expressing (PV) interneurons through the transcription factor SMAD1, which controls an array of glutamatergic synapse proteins and components of perineuronal nets. PV-interneuron-specific disruption of BMP2–SMAD1 signalling is accompanied by a loss of glutamatergic innervation in PV cells, underdeveloped perineuronal nets and decreased excitability. Ultimately, this impairment of the functional recruitment of PV interneurons disrupts the cortical excitation–inhibition balance, with mice exhibiting spontaneous epileptic seizures. Our findings suggest that developmental morphogen signalling is repurposed to stabilize cortical networks in the adult mammalian brain.

Neuronal circuits in the neocortex underlie our ability to perceive our surroundings, integrate various forms of sensory information and support cognitive functions. Cortical computation relies on assemblies of excitatory and inhibitory neuron types that are joined into canonical microcircuit motifs. The synaptic innervation and intrinsic properties of fast-spiking parvalbumin-expressing inhibitory interneurons (PV interneurons) have emerged as key parameters for controlling cortical circuit stability and plasticity[1,6]. During development, sensory experience shapes the synaptic innervation of PV interneurons in an afferent-specific manner, and synaptic input to PV interneuron dendrites is a crucial node for cortical dysfunction in disorders[7–11]. In the adult brain, neuronal-activity-dependent regulation of the recruitment and excitability of PV interneurons is fundamental for maintaining the balance between excitation and inhibition, and has been implicated in gating cortical circuit plasticity during learning processes[1,2,12–15]. However, the molecular mechanisms that underlie these features—in particular, the transcellular signalling events that relay alterations in neuronal network activity and adjust PV interneuron function—are poorly understood.

## Neuronal network activity mobilizes BMP signalling

To identify candidate transcellular signals that are regulated by neuronal network activity in mature neocortical neurons, we examined secreted growth factors of the bone morphogenetic protein (BMP) family, which have been implicated in cell-fate specification and neuronal growth during development[16–24]. We examined four bone morphogenetic proteins (BMP2, BMP4, BMP6 and BMP7) in mice, and found that *Bmp2* mRNA was significantly upregulated in glutamatergic neurons after stimulation (3.5 ± 0.5-fold; Extended Data Fig. 1a–d). A similar activity-dependent increase in BMP2 was observed at the protein level in neurons derived from a *Bmp2* HA-tag knock-in mouse (*Bmp2^HA/HA*; Extended Data Fig. 1e–g and Supplementary Information). As developmental morphogens, BMPs direct gene regulation in recipient cells through SMAD transcription factors[25–29] (Fig. 1a). Notably, the canonical BMP target genes *Id1* and *Id3* were significantly upregulated in stimulated neocortical cultures, and this process was blocked by the addition of the extracellular BMP antagonist Noggin (Extended Data Fig. 1h,i). In the neocortex of adult mice, key BMP signalling components continue to be expressed, with the ligand BMP2 exhibiting the highest mRNA levels in glutamatergic neurons (Extended Data Fig. 2a–c). To test whether the transcription of BMP target genes is activated in response to increased neuronal network activity in adult mice, we chemogenetically silenced upper-layer PV interneurons in the barrel cortex (Fig. 1b). This local reduction of PV-neuron-mediated inhibition results in increased neuronal network activity[30,31] accompanied by a four- to eightfold transcript increase in the activity-induced primary response genes *Fos* and *Bdnf* (Fig. 1c). Of note, this chemogenetic stimulation also

[1]Biozentrum, University of Basel, Basel, Switzerland. [2]Swiss Institute of Bioinformatics, Basel, Switzerland. [3]Friedrich Miescher Institute for Biomedical Research, Basel, Switzerland. [4]Present address: Department of Biomedicine, University of Basel, Basel, Switzerland. [5]Present address: Roche Pharmaceutical Research and Early Development, Roche Innovation Center Basel, Basel, Switzerland. ✉e-mail: peter.scheiffele@unibas.ch

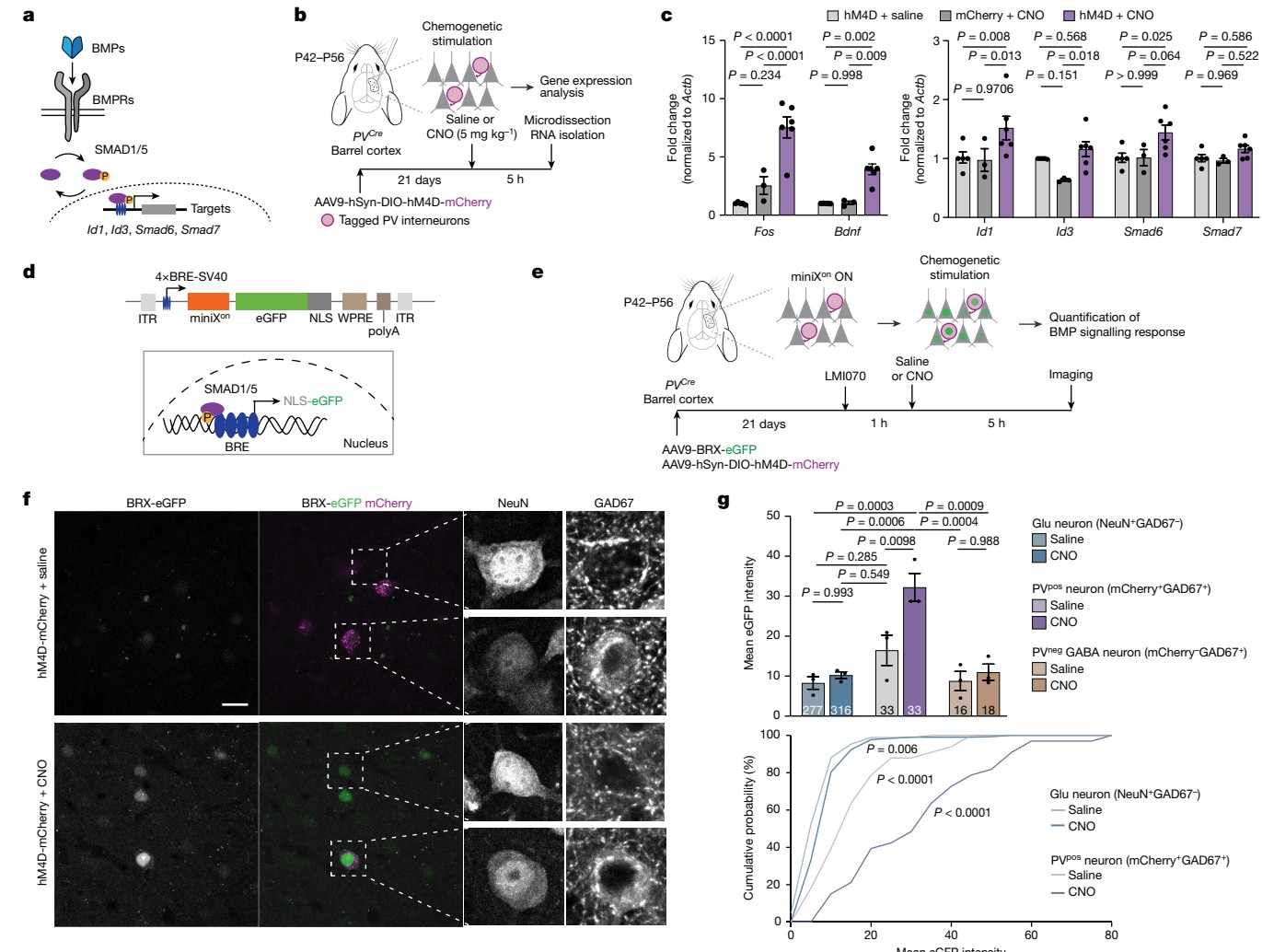

**Fig. 1 | Increased neuronal activity elicits BMP signalling in PV interneurons of the adult barrel cortex. a**, Illustration of BMP pathway components. BMPRs, BMP receptors. **b**, Schematic representation of the protocol for chemogenetic manipulation of neuronal activity in the adult barrel cortex. P42, postnatal day 42; P56, postnatal day 56. **c**, Expression of the immediate early genes *Fos* and *Bdnf* and the SMAD1/5 target genes *Id1*, *Id3*, *Smad6* and *Smad7* in the barrel cortex of chemogenetically stimulated and control mice (*n* = 3–6 mice per group, mean ± s.e.m., two-way ANOVA with Tukey's post-hoc test). **d**, Schematic representation of the viral BRX reporter. Nucleus-targeted eGFP (NLS-eGFP) is expressed under the control of regulatory elements from the *Id1* gene (4×BRE),

a minimal SV40 promoter and the miniX[on] splicing cassette. ITR, inverted terminal repeat. **e**, Experimental paradigm. **f**, Representative images of the BRX reporter signal in barrel cortex layer 2/3 of *PV^Cre* mice. Cre-dependent mCherry identifies PV cells, NeuN identifies neurons and the somatic–perinuclear GAD67 signal identifies GABA neurons. Scale bar, 20 μm. **g**, BRX reporter-driven nuclear eGFP intensity per mouse (*n* = 3 mice per group, cell numbers indicated in columns, mean ± s.e.m., one-way ANOVA with Tukey's multiple comparisons) and cumulative distribution of eGFP reporter intensity per cell for glutamatergic and PV-positive neurons (Kolmogorov–Smirnov test).

resulted in the upregulation of BMP target genes (*Id1 and Smad6*, and an increase in *Id3* when compared with mCherry + clozapine *N*-oxide (CNO) negative controls) (Fig. 1c). We then mapped neuronal cell populations in which BMP target genes were activated in response to neuronal network activity, using a novel temporally controlled BMP signalling reporter (BMP-responsive X^on; BRX) (Fig. 1d). We combined BMP-response element sequences (4×BRE) from the *Id1* promoter[32] with the small molecule (LMI070)-gated miniX^on cassette[33] to drive a nucleus-targeted eGFP (Extended Data Fig. 3). Thus, the level of nuclear eGFP reports the activation of BMP signalling during a time window specified by LMI070 application (Extended Data Fig. 4a–f). Chemogenetic stimulation resulted in a selective increase in the activity of the BRX reporter in PV interneurons, whereas the mean reporter output in glutamatergic cells and non-PV interneurons was unchanged (Fig. 1f,g, but note that a sparse subpopulation of NeuN⁺Gad67⁻ glutamatergic neurons did show a significant reporter signal). Genetic restriction of the BRX reporter to PV interneurons revealed a threefold increase in

the BRX signal in response to chemogenetic stimulation (Extended Data Fig. 4g–i). Together, these results show that increased cortical network activity mobilizes BMP2 and selectively activates the BMP signalling pathway in PV interneurons in the barrel cortex of adult mice.

## BMP–SMAD1 signalling regulates synaptic proteins

During development, the combinatorial action of various BMP ligands and receptors directs the cell-type-specific regulation of target genes through SMAD transcription factors, but SMAD-independent functions have also been described[16,20,22,34–36]. In neocortical neurons, stimulation with BMP2 (20 ng ml⁻¹ for 45 min) resulted in the activation of SMAD1 and SMAD5 (hereafter, SMAD1/5) in both glutamatergic and GABAergic (γ-aminobutyric-acid-producing) neurons (Extended Data Fig. 5a–c). To uncover neuronal SMAD1 target genes, we performed chromatin immunoprecipitation followed by sequencing (ChIP–seq) for SMAD1/5 in adult mouse neocortex and neocortical cultures

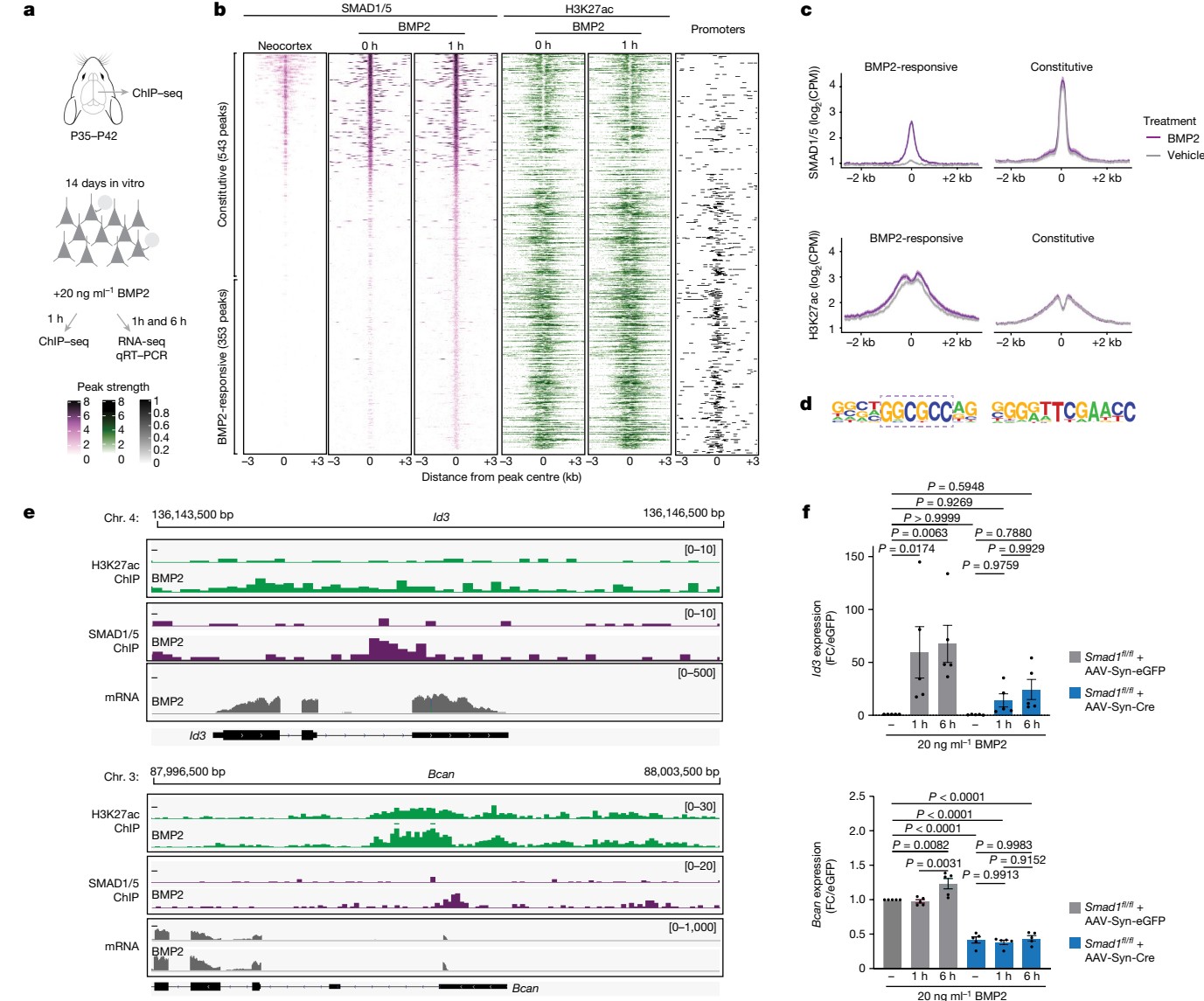

**Fig. 2 | Neuronal BMP2–SMAD1 signalling regulates synaptic components.**
**a**, Schematic representation of ChIP–seq and RNA-seq experiments from mouse neocortex and neocortical cultures. qRT–PCR, quantitative PCR with reverse transcription. **b**, ChIP–seq analysis of neocortical tissue and naive (0 h) or growth-factor-stimulated (1 h 20 ng ml⁻¹ BMP2) neocortical neuron cultures at DIV14 (14 days in vitro). Heat maps in purple show the peak strength of SMAD1/5 binding; heat maps in green show H3K27ac binding at SMAD1/5 peak regions. The right column (in black) shows the position of promoter elements. Each binding site is represented as a single horizontal line centred at the SMAD1/5 peak summit; the colour intensity correlates with the sequencing signal for the indicated factor. Peaks are ordered by decreasing SMAD1/5 peak intensity. **c**, Mean normalized ChIP–seq signal for SMAD1/5 and H3K27ac plotted for BMP2-responsive and constitutive SMAD1/5-binding sites. Grey lines indicate signal obtained from vehicle-treated cultures and purple lines indicate signal from BMP2-stimulated cultures. **d**, Top enriched motifs detected for BMP2-responsive (left) and constitutive (right) SMAD1/5 peaks. **e**, Examples of IGV genome browser ChIP–seq tracks showing the H3K27ac (green), SMAD1/5 (purple) and RNA-seq (grey) signals for the SMAD1/5 targets *Id3* and *Bcan* in naive (−) and BMP2-stimulated cultures. **f**, qPCR analysis of the mRNA expression of *Id3* and *Bcan* in AAV-Syn-eGFP infected versus AAV-Syn-Cre infected *Smad1^{fl/fl}* neocortical cultured neurons. Fold change (FC) relative to unstimulated cells is shown for 1 h and 6 h stimulation with 20 ng ml⁻¹ BMP2. Bar graphs show mean ± s.e.m. ($n = 5$ independent cultures per condition, one-way ANOVA with Tukey's multiple comparisons).

(Fig. 2a). We identified 239 and 543 sites that were bound in the mouse neocortex and in cultured neocortical neurons, respectively (Fig. 2b and Supplementary Table 1). Notably, 77% of the binding sites in vivo were reproduced in the cultured neuron preparations. To specifically map sites that are acutely regulated by BMP–SMAD1/5 signalling, we stimulated cortical cultures by adding recombinant BMP2. After stimulation, we identified another 353 BMP2-responsive SMAD1/5-binding sites. Most of the BMP2-responsive peaks were associated with promoter elements. To investigate whether SMAD1/5 trigger the de novo activation of target genes or, rather, modifies the transcriptional output of active genes, we mapped histone H3 acetylated at lysine 27 (H3K27ac)

marks, a chromatin modification at active promoters and enhancers. By intersecting H3K27ac ChIP–seq signals with SMAD1/5 peaks (Fig. 2b,c), we found that most BMP2-responsive elements contain significant H3K27ac marks, which are slightly increased after stimulation. This suggests that many of these sites are already active without BMP2 stimulation. By comparison, constitutively bound regions exhibited a lower H3K27ac signal (Fig. 2b,c). Sequence analysis identified an enrichment of different motifs for SMAD1/5 DNA binding in the constitutive (tissue and neuronal culture) and in the BMP2-responsive gene-regulatory elements, suggesting that DNA binding involves different co-factors (Fig. 2d). The effect of the BMP2-induced recruitment of SMAD1/5 on

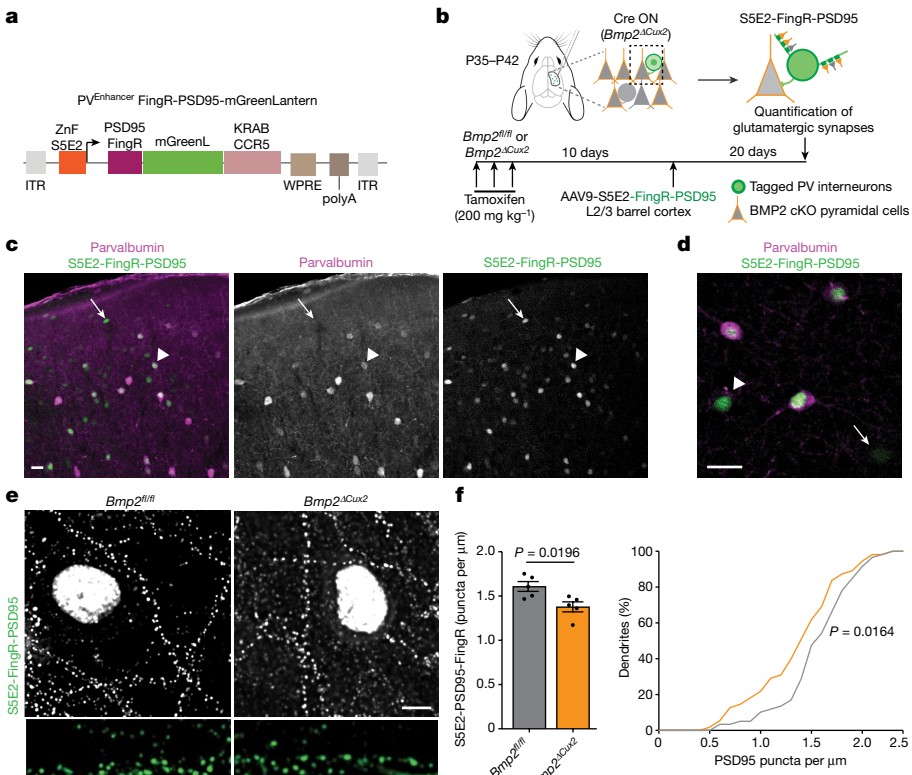

**Fig. 3 | Pyramidal-cell-derived BMP2 regulates the innervation of PV interneurons. a**, Schematic representation of the viral vector for expression of the glutamatergic FingR-PSD95 probe. FingR expression is driven from an S5E2 PV enhancer fused to a CCR5 zinc-finger-binding site (ZnF). The FingR coding sequence is fused to mGreenLantern and a CCR5-KRAB transcriptional repressor for autoregulation of probe expression. Thus, excess probe accumulates in the nucleus and reduces probe expression. **b**, Conditional deletion of BMP2 from upper-layer neurons in *BMP2*$^{\Delta Cux2}$ mice (*Cux2*$^{creERT2}$::*Bmp2*$^{fl/fl}$) is achieved by three applications of tamoxifen spaced over one week. cKO, conditional knockout. **c**, Selectivity of FingR probe expression with an S5E2 enhancer. A PV interneuron with a modest level of parvalbumin protein is marked with an arrowhead and a parvalbumin-negative cell is marked with an arrow. **d**, Higher magnification view of cells marked as in **c**. Scale bars, 20 µm. **e**, FingR-PSD95-marked synapses formed onto PV interneurons in control (*Bmp2*$^{fl/fl}$) and *Bmp2*$^{\Delta Cux2}$ mice and corresponding dendritic stretches. Scale bars, 5 µm. **f**, Left, quantification of the density of glutamatergic synapses on the dendrites of PV interneurons (identified by probe expression and parvalbumin immunostaining). The number of synapses was normalized to the dendritic length (mean ± s.e.m. from *n* = 5 mice per genotype, *n* = 7–17 dendrites per mouse, unpaired two-tailed *t*-test). Right, cumulative distribution of synapse density across all dendrites analysed (*n* = 55–60 dendrites, Kolmogorov–Smirnov test).

transcriptional output was examined by RNA sequencing (RNA-seq). Differential gene expression analysis identified 30 and 147 upregulated transcripts 1 h and 6 h after BMP2 stimulation, respectively (Extended Data Fig. 5d and Supplementary Table 2). Fifty per cent of the regulated genes 1 h after BMP2 stimulation had direct SMAD1/5 binding at their promoters. These genes included known negative-feedback-loop genes of the BMP signalling pathway (*Id1*, *Id3* and *Smad7*). Twenty-five per cent of differentially regulated genes 6 h after BMP2 stimulation had direct SMAD1/5 binding (Extended Data Fig. 5d). Conditional knockout of *Smad1* in postmitotic neurons was sufficient to abolish the upregulation of these genes in response to BMP2 signalling and reduce their expression in naive (unstimulated) neurons (Fig. 2f, Extended Data Fig. 5e,f and Supplementary Table 3). Direct transcriptional targets of BMP–SMAD1 signalling in neocortical neurons included an array of activity-regulated genes such as *Junb*, *Trib1* and *Pim3*, as well as genes that encode key components of the extracellular matrix (*Bcan* and *Gpc6*) and glutamatergic synapses (*Lrrc4* and *Grin3a*) (Fig. 2e and Extended Data Fig. 5g,h). Moreover, neuronal ablation of *Smad1* was accompanied by broad changes in gene expression beyond the deregulation of direct SMAD1 target genes (Extended Data Fig. 5i). Top gene ontology (GO) terms enriched amongst the upregulated genes were 'glutamatergic synapse' and transcription factors under the term 'nucleus' (Extended Data Fig. 5j). Furthermore, deregulated genes included the majority of neuronal-activity-regulated rapid primary response genes (rPRGs) and

secondary response genes (SRGs) (Extended Data Fig. 5k). Thus, SMAD1 is a key downstream mediator of BMP signalling in mature neurons and its neuronal loss of function results in a substantial upregulation of neuronal activity response genes in vitro.

## SMAD1 controls the innervation of PV interneurons

In neocortical circuits, the excitation–inhibition balance is regulated by glutamatergic input synapses onto PV interneurons, and perineuronal nets (PNNs) surrounding these cells are modified in response to changes in neuronal network activity[37,38]. To test whether pyramidal-cell-derived BMP2 modifies the innervation of PV interneurons, we generated *Bmp2* conditional knockout mice in which *Bmp2* is selectively ablated in upper-layer glutamatergic neurons (*Cux2*$^{creERT2}$::*Bmp2*$^{fl/fl}$; referred to as *Bmp2*$^{\Delta Cux2}$ mice). We then adopted genetically encoded intrabodies (fibronectin intrabodies generated by mRNA display; FingRs) to quantitatively map the synaptic inputs to PV interneurons[39,40] (Extended Data Fig. 6a–c and Supplementary Video 1). A FingR-PSD95 probe was selectively expressed in PV interneurons in layer 2/3 of the barrel cortex under the control of a PV-cell-specific enhancer[41] (Fig. 3a–d). Notably, the density of synapses onto PV interneurons was reduced after genetic ablation of *Bmp2* in upper-layer pyramidal cells of *Bmp2*$^{\Delta Cux2}$ mice (Fig. 3e,f). We then generated PV-interneuron-specific *Smad1* conditional knockout mice to examine whether BMP2 acts through

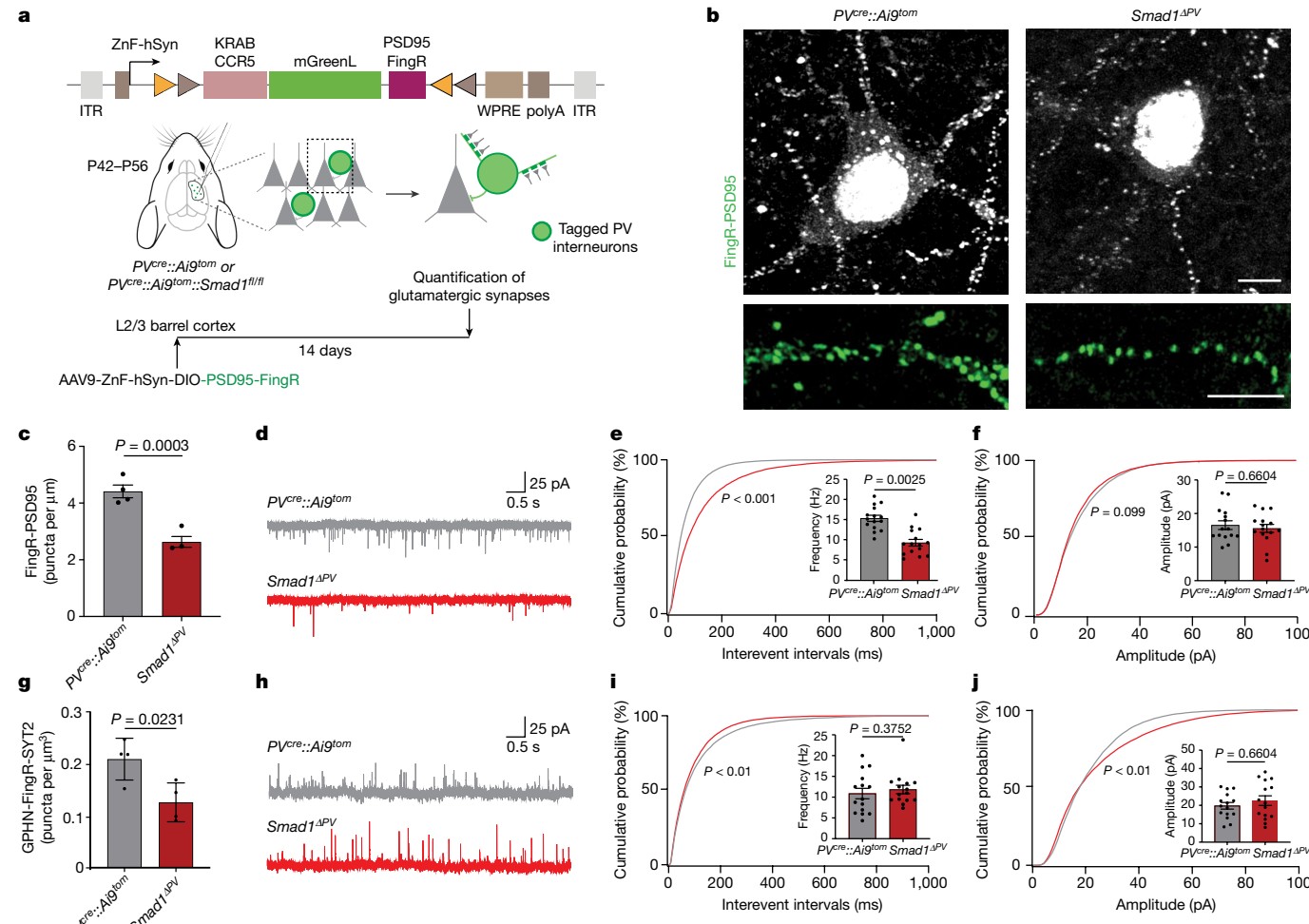

**Fig. 4 | SMAD1 regulates the glutamatergic innervation of PV interneurons.**
**a**, Schematic representation of the Cre-recombinase-dependent intrabody probe. Intrabody expression is driven from the human synapsin promoter (hSyn) fused to a CCR5 ZnF. **b**, FingR-PSD95-marked synapses formed onto control ($PV^{cre}::Ai9^{tom}$) and Smad1 cKO ($Smad1^{\Delta PV}$) PV interneurons and corresponding dendritic stretches. Scale bars, 5 μm. **c**, Quantification of the density of glutamatergic synapses on the dendrites of PV interneurons. The number of synapses was normalized to the dendritic length (mean ± s.e.m. from $n$ = 3 or 4 mice per genotype, $n$ = 10 dendrites per mouse, unpaired two-tailed $t$-test). Note that the vast majority of FingR-PSD95-marked structures co-localize with the presynaptic marker vGluT1 (see Extended Data Fig. 6a). **d**, Representative traces of mEPSC recordings from control (grey) and $Smad1^{\Delta PV}$ (red) PV interneurons in acute slice preparations from adult mice. **e**, Frequency distribution of interevent intervals (Kolmogorov–Smirnov test) and mean mEPSC frequency (mean ± s.e.m. for $n$ = 15 cells per genotype, from $n$ = 3 or

4 mice, Kolmogorov–Smirnov test). **f**, Frequency distribution of mEPSC amplitudes (Kolmogorov–Smirnov test) and mean mEPSC amplitude (mean ± s.e.m. for $n$ = 15 cells per genotype, from $n$ = 3 or 4 mice, Kolmogorov–Smirnov test). **g**, Quantification of PV–PV GABAergic synapse density on PV interneuron somata. The number of GPHN-FingR-eGFP and synaptotagmin-2 (SYT2)-containing structures was normalized to the soma volume (see Extended Data Figs. 6d,e and 7d,e for details) (mean ± s.e.m. from $n$ = 3 or 4 mice per genotype, $n$ = 78 cells, unpaired two-tailed $t$-test). **h**, Representative traces of mIPSCs recorded from control (grey) and $Smad1^{\Delta PV}$ (red) PV interneurons. **i**, Frequency distribution of interevent intervals (Kolmogorov–Smirnov test) and mean mIPSC frequency (mean ± s.e.m. for $n$ = 15 cells per genotype, from $n$ = 3 or 4 mice, Kolmogorov–Smirnov test). **j**, Frequency distribution of mIPSC amplitudes (Kolmogorov–Smirnov test) and mean mIPSC amplitude (mean ± s.e.m. for $n$ = 15 cells per genotype, from $n$ = 3 or 4 mice, Kolmogorov–Smirnov test).

SMAD1. Postnatal ablation of *Smad1* ($PV^{cre/+}::Smad1^{fl/fl}$; referred to as $Smad1^{\Delta PV}$ mice) did not alter the density or distribution of PV cells in the somatosensory cortex (Extended Data Fig. 7a–c). Using a Cre-recombinase-dependent form of the FingR-PSD95 probes (Fig. 4a), we observed a 40% reduction in the density of glutamatergic synapses as observed by morphology onto $Smad1^{\Delta PV}$ interneurons (Fig. 4b,c). This was accompanied by a comparable reduction in the frequency of miniature excitatory postsynaptic currents (mEPSCs), but there was no change in mEPSC amplitude in acute slice recordings (Fig. 4d–f). The density of perisomatic PV–PV synapses (identified by synaptotagmin-2 and a FingR-gephyrin probe[39]) was also reduced (Fig. 4g and Extended Data Fig. 7d,e). However, there was no significant change in the frequency or amplitude of miniature inhibitory postsynaptic currents

(mIPSCs) in PV cells of $Smad1^{\Delta PV}$ mice, owing probably to compensatory inhibition derived from other interneuron classes (Fig. 4h–j). Thus, SMAD1 is required for normal functional glutamatergic innervation of layer 2/3 PV interneurons, and the loss of SMAD1 results in reduced glutamatergic input to these cells in $Smad1^{\Delta PV}$ mice.

Neuronal-activity-induced regulation in PV interneurons modifies the elaboration of PNNs and parvalbumin expression[6,30,37,38], and our ChIP–seq analysis identified the PNN component brevican (*Bcan*) as one of the direct SMAD1 targets. In $Smad1^{\Delta PV}$ mice, the elaboration of PNNs around PV interneurons and the expression of parvalbumin protein were significantly reduced (Fig. 5a–c and Extended Data Fig. 8a–d). Conversely, PV-cell-specific activation of the BMP signalling pathway by expression of a constitutively active BMP receptor was sufficient to

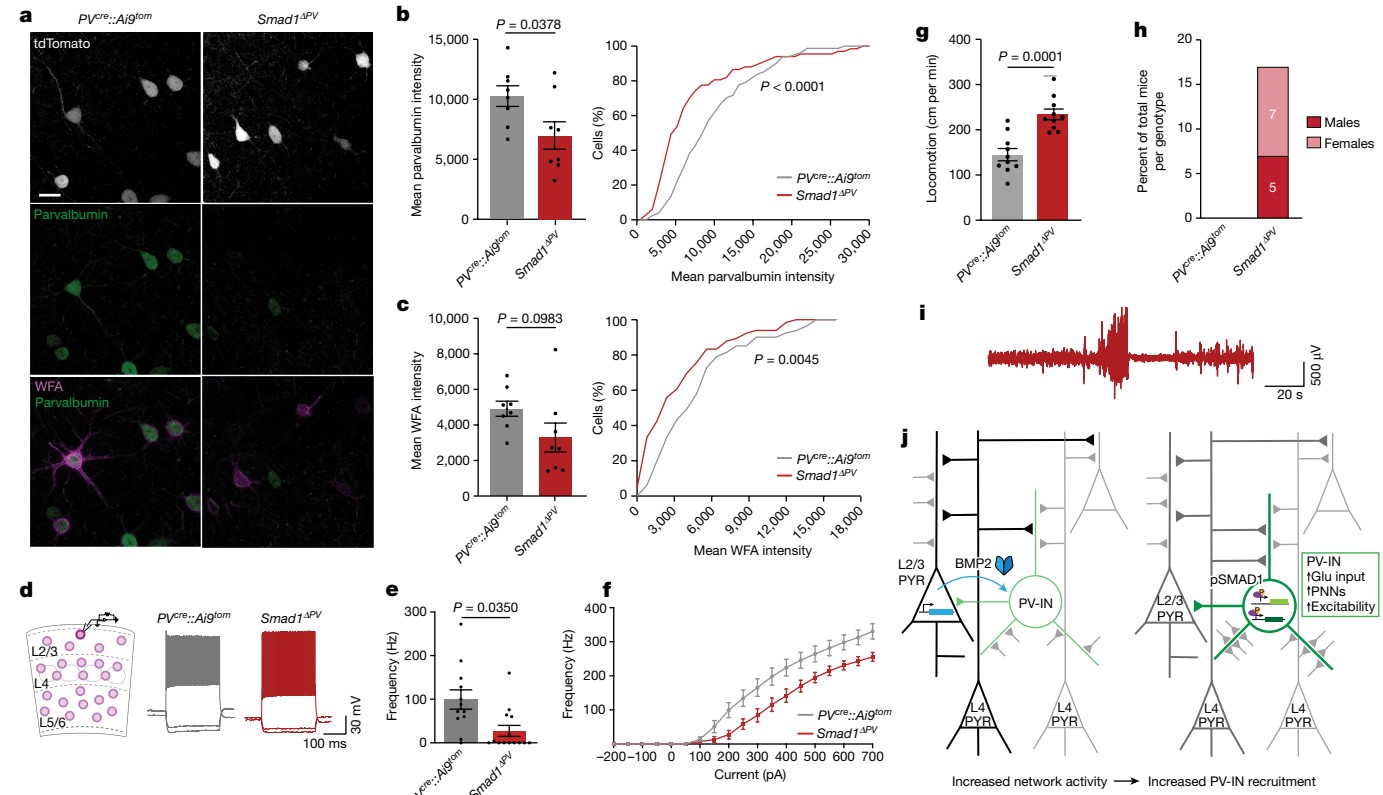

**Fig. 5 | Loss of SMAD1 in PV interneurons results in disruption of the excitation–inhibition balance in adult mice. a**, Parvalbumin immunoreactivity and *Wisteria floribunda* agglutinin (WFA) binding to PNNs in adult control (*PV^cre::Ai9^tom*) and *Smad1* cKO (*Smad1^ΔPV*) mice (postnatal day (P)56–P72). Scale bar, 20 μm. **b**, Quantification of parvalbumin immunoreactivity per cell in *PV^cre::Ai9^tom* (grey) and *Smad1^ΔPV* (red) mice. Bar graphs with mean intensity per mouse (*n* = 8 per genotype) and cumulative distribution of mean intensity per cell (*n* = 70 cells for *PV^cre::Ai9^tom*; *n* = 63 cells for *Smad1^ΔPV* mice). Unpaired two-tailed *t*-test for bar graph and Kolmogorov–Smirnov test for cumulative distribution. **c**, WFA staining intensity plotted as in **b**. **d**, Example traces from current-clamp recordings of control (grey) and *Smad1^ΔPV* (red) PV interneurons (layer 2/3; P56–P72) in acute slice preparations from adult mice. **e,f**, Mean firing frequency in response to 200 pA current injection (**e**) and comparison of firing frequencies of layer 2/3 PV interneurons at given currents (**f**) in cells from *PV^cre::Ai9^tom* (grey) and *Smad1^ΔPV* (red) mice (*n* = 4 mice, *n* = 12 cells for *PV^cre::Ai9^tom* and *n* = 4 mice, *n* = 14 cells for *Smad1^ΔPV*, Kolmogorov–Smirnov test). **g**, Quantification of locomotion in the open field from *PV^cre::Ai9^tom* (grey) and *Smad1^ΔPV* (red) mice (9–13 weeks of age, *n* = 10 mice per genotype, unpaired two-tailed *t*-test). **h**, Number of *PV^cre::Ai9^tom* control (0 out of 65 mice) and male and female *Smad1^ΔPV* (red) mice (12 out of 71 mice) showing spontaneous seizures during cage changes. Note that the observation of spontaneous seizures was an exclusion criterion for the morphological, electrophysiological and molecular analyses performed in this study. **i**, Representative 2.5-min EEG trace obtained from a *Smad1^ΔPV* mouse in long-term monitoring (continuous recording for three weeks). All bar graphs show mean ± s.e.m. **j**, Working model of transcellular BMP2–SMAD1 signalling in cortical microcircuits. pSMAD1, phosphorylated SMAD1; PV-IN, PV interneuron; PYR, pyramidal cell.

increase the levels of parvalbumin (Extended Data Fig. 8e–g, but note that parvalbumin was not identified as a SMAD1 target in ChIP experiments; Supplementary Table 1). Through organizing PNNs, brevican has been implicated in regulating the plasticity and excitability of PV interneurons[38]. Notably, the firing rate of SMAD1-deficient PV interneurons in response to current injections was significantly reduced in the barrel cortex of adult mice (Fig. 5d–f and Extended Data Fig. 9a, note that the firing rate as well as the mEPSC frequency was unchanged in young mice; Extended Data Fig. 9b–e). This reduced firing frequency is most likely to be explained by a reduction in input resistance in the *Smad1^ΔPV* cells (Extended Data Fig. 9a). Thus, in the absence of BMP–SMAD1 signalling, PV interneurons not only receive less glutamatergic drive, but they are also less excitable. These cellular alterations resulted in a severe overall disruption of the cortical excitation–inhibition balance. Compared with control littermates, *Smad1^ΔPV* mice exhibited hyperactivity in open-field tests and frequently exhibited spontaneous seizures when introduced into novel environments (Fig. 5g,h). Video-coupled long-term electroencephalogram (EEG) recordings (three weeks of continuous monitoring) with electrodes over the barrel cortex (Supplementary Video 2) revealed marked high-amplitude bursts of activity at the time of seizure, followed by a refractory period

(Fig. 5i). Overall, our results show that increased network activity in the somatosensory cortex of adult mice triggers the upregulation of BMP2 in glutamatergic neurons, which balances excitation by controlling the synaptic innervation and function of PV interneurons through the transcription factor SMAD1 (Fig. 5j).

## Discussion

Despite being exposed to a wide range of sensory stimulus intensities, cortical circuits exhibit remarkably stable activity patterns that enable optimal information coding by the network. This network stability is achieved by homeostatic adaptations that modify the excitability of individual neurons and scale the strength of synapses, as well as by microcircuit-wide modifications of the density of excitatory and inhibitory synapses[2,15,42–44]. These adaptations happen at various timescales, from near instantaneous adjustments of excitation and inhibition during sensory processing[45], to slower modifications of synaptic connectivity after longer-term shifts in circuit activation as they occur during sensory deprivation but also in disease states[3–5,15,46,47]. Thus, both rapid cell-intrinsic and long-lasting transcellular signalling processes have evolved to ensure the function and stability of the cortical network.

We show here that increased neuronal network activity in the somatosensory cortex of adult mice triggers the upregulation of BMP2 in pyramidal cells and the expression of BMP target genes in PV interneurons. We hypothesize that this rise in activity not only increases the expression of BMP2 at the level of transcription (Extended Data Fig. 1), but is also likely to promote its release from dense core vesicles[48]. Direct testing of this hypothesis in the mouse neocortex will require better tools for visualizing endogenous BMP2. The transcription factor SMAD1 directly binds to and regulates the promoters of an array of glutamatergic synapse proteins, ion channels and components of the PNNs. Further studies will be needed to define specific contributions of individual SMAD1 target genes in PV interneurons. However, our genetic analysis shows that BMP2–SMAD signalling provides a transneuronal signal to adjust the innervation and excitability of PV interneurons, which ultimately serves to maintain the excitation–inhibition balance and stabilize cortical network function in the adult neocortex. Notably, the SMAD1 loss-of-function phenotypes only emerge with age, as the excitability and synaptic innervation of PV interneurons are normal in juvenile (P26–P30) mice. In the developing auditory cortex, genetic deletion of type I BMP receptors from PV interneurons is associated with impaired synaptic plasticity at the output synapses of PV interneurons onto principal neurons of layer 4, whereas basal GABAergic transmission remains unchanged[49]. This suggests that BMP2–SMAD1 signalling has a selective role in controlling glutamatergic input connectivity to PV interneurons.

Notably, transcriptional regulation through BMP2–SMAD1 signalling differs considerably from the action of activity-induced immediate early genes. As a secreted growth factor, BMP2 derived from glutamatergic neurons relays high network activity to PV interneurons through the activation of an array of SMAD1 target genes. Rather than ON/OFF responses, most direct SMAD1 targets exhibit active enhancer and promoter elements and are already expressed under basal conditions. However, SMAD1 activation results in an increase of transcriptional output, indicating a graded gene-expression response to BMP2.

In early development, BMP growth factors act as morphogens that carry positional information and differentially instruct cell fates[26,27,29]. The combinatorial complexity arising from the substantial number of BMP ligands and receptors has the power to encode computations for finely tuned cell-type-specific responses[34,50]. Our work suggests that the spatio-temporal coding power, robustness and flexibility that evolved for developmental patterning are harnessed for balancing the plasticity and stability of neuronal circuits in the adult mammalian brain. Of note, other BMP ligands besides BMP2 are selectively expressed in neocortical cell types (Extended Data Fig. 2b). Moreover, an array of type I and type II BMP receptors are detected across neocortical cell populations[51]. Thus, BMP–SMAD1 signalling might control other aspects of neuronal cell–cell communication.

Disruptions in the excitation–inhibition balance and homeostatic adaptations have been implicated in neurodevelopmental disorders, and reduced GABAergic signalling and a propensity to develop epilepsy are often seen in individuals with autism[3–5,52]. Considering that BMP signalling pathways can be targeted with peptide mimetics[53], these might provide an entry point for therapeutic interventions in neurodevelopmental disorders that are characterized by disruptions in the innervation of PV interneurons, the excitation–inhibition balance and seizures.

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

## Methods

### Mice

All procedures involving animals were approved by and performed in accordance with the guidelines of the Kantonales Veterinäramt Basel-Stadt. All experiments were performed in mice on a C57Bl/6J background, except for some of the experiments performed in cultured wild-type neurons, which used RjOrl:SWISS mice (Janvier). All mice were group housed (weaning at P21–P23) under a 12-h light–dark cycle (06:00–18:00) at 21–24 °C and 50–60% humidity with food and water ad libitum. Both males and females were used at similar numbers for the experiments. Mice were randomly assigned to treatment groups. Mice that exhibited a spontaneous seizure were excluded from molecular, anatomical and slice physiology analyses.

Smad1[fl/fl] mice[54], Pvalb-cre mice[55] and Ai9 mice[56] were obtained from Jackson Laboratories (Jax stock no: 008366, 017320 and 007909, respectively). Cux2-CreERT2 mice[57] were obtained from the Mutant Mouse Resource and Research Center (MMRRC). Bmp2-2xHA mice were generated using a CRISPR–Cas9 strategy[58] inserting a double HA tag at the N terminus of the mature BMP2 protein, between amino acids S292 and S293. The guide RNAs (gRNAs) used were 5′-GTCTCTTGCAGCTGGACTT G-3′ and 5′-CAAAGGGTGTCTCTTGCAGC-3′, together with a 200-bp single-stranded DNA ultramer: 5′-GACTTTTGGACATGATGGAAAAGGA CATCCGCTCCACAAACGAGAAAAGCGTCAAGCCAAACACAAACAGCGG AAGCGCCTCAAGTCCGCTAGC**TACCCATACGATGTTCCAGATTACGCT** GGCTATCCCTATGACGTCCCGGACTATGCAGCTAGCAGCTGCAAGAGAC ACCCTTTGTATGTGGACTTCAGTGATGTGG-3′ (the sequence encoding the HA tags is highlighted in bold).

### Surgery and drug treatments

Injections of recombinant AAVs were performed into the barrel cortex of 42–49-day-old male and female mice performed under isoflurane anaesthesia (Baxter). Mice were placed in a stereotactic frame (Kopf) and a small incision (0.5–1 cm) was made over the barrel cortex at the following coordinates targeting two sites: mediolateral (ML) ±3.0 mm and ±3.4 mm, at anteroposterior (AP) 0.6 mm and AP −1.6 mm, dorsoventral (DV) −1.5 mm from Bregma to target layers 2/3 and 4. For injections of FingR intrabodies, two injection sites restricted to layer 2/3 were used: ML ±3.0 mm and ±3.4 mm at AP −1.0 mm, DV −0.96 mm from Bregma. Recombinant AAVs (titre: $10^{12}$–$10^{13}$) were injected via a glass capillary with an outer diameter of 1 mm and an inner diameter of 0.25 mm (Hilgenberg) for a total volume of 100 nl per injection site. The wound was closed with sutures (Braun, C0766194).

LMI070 (25 mg kg$^{-1}$, MedChemExpress, HY-19620, suspended in 20% cyclodextrin and 10% dimethyl sulfoxide (DMSO) to 5 mg ml$^{-1}$ concentration) was administrated by oral gavage. Clozapine N-oxide (CNO) (5 mg kg$^{-1}$, Sigma Aldrich, C0832) and doxycycline (50 mg kg$^{-1}$, Thermo Fisher Scientific, BP26531, suspended in 0.9% NaCl to 5 mg ml$^{-1}$ concentration) were administered by intraperitoneal injection.

### Antibodies and probes

Primary antibodies were: monoclonal mouse anti-synaptotagmin-2 (Zebrafish International Resource Center, ZNP-1), rabbit anti-SMAD1 (Cell Signaling 6944, 1:100 for ChIP and 1:1,000 for western blot), H3K27ac (Abcam 4729, 1:1,000), rabbit anti-SMAD5 (Cell Signaling, 12534, 1:100 for ChIP and 1:1,000 for western blot), anti-phospho-SMAD1/5/9 (Cell Signaling 13820, 1:1,000), mouse anti-BMPR2 (BD Pharmingen, 612292, 1:1,000), rabbit anti-calnexin (StressGen, SPA-865, 1:2,000), mouse anti-MAP2 (Synaptic Systems, 188011, 1:1,000), mouse anti-CAMKII alpha (Thermo Fisher Scientific, MA1-048, 1:800), rat anti-GAPDH (Biolegend, 607902, 1:10,000), rabbit anti-NeuN (Abcam, ab177487, 1:500), mouse anti-GAD67 (Millipore MAB5406, 1:500), rabbit anti-vGLUT1 (Synaptic Systems 135303, 1:5,000), biotinylated WFA (Vector Laboratories B-1355-2, 1:500), rabbit anti-HA (Cell Signaling 3724, 1:1,000), mouse anti-GFP antibody (Santa Cruz, sc-9996, 1:1,000) and goat anti-parvalbumin antibody (Swant PVG213, 1:5,000). Secondary antibodies were: HRP goat anti-rabbit (Jackson 111-035-003, 1:10,000), HRP goat anti-rat (Jackson 112-035-143, 1:10,000), HRP goat anti-mouse (Jackson 115-035-149, 1:10,000), Alexa405 goat anti-rabbit (Thermo Fisher Scientific A-31556, 1:500), Alexa488 donkey anti-rabbit (Thermo Fisher Scientific R37118, 1:1,000), Alexa647 donkey anti-mouse (Jackson 715-605-151, 1:1,000), Alexa647 streptavidin (Thermo Fisher Scientific, S32357, 1:1,000), Cy2 Streptavidin (Jackson 016-220-084, 1:1,000), Cy3 donkey anti-mouse (Jackson 715-165-151, 1:500), Cy3 donkey anti-rabbit (Jackson 711-165-152, 1:500), Cy5 donkey anti-goat (Jackson 705-175-147, 1:500), Cy5 donkey anti-rabbit (Jackson 711-175-152, 1:500) and Cy5 donkey anti-mouse (Jackson 715-175-511, 1:500). DAPI dye was used for nuclear staining (TOCRIS Bio-Techne, 5748, 1:5,000).

### Immunohistochemistry and image acquisition

Mice were deeply anaesthetized with a ketamine–xylazine mix (100 and 10 mg per kg, respectively, intraperitoneally) and were transcardially perfused with fixative (4% paraformaldehyde (PFA) in 0.1 M phosphate buffer, pH 7.4). For synapse quantifications with FingR probes the fixative also contained 15% picric acid. After perfusion, brains were post-fixed overnight in fixative at 4 °C and washed three times with 100 mM phosphate buffer.

For quantifications of parvalbumin and WFA expression and BRX reporter analyses, coronal brain slices were cut at 40 μm with a Vibratome (VT1000S, Leica). For FingR-PSD95 analysis with the Cre-dependent reporter, coronal brain slices were cut at 30 μm with a Cryostat (Microm HM560, Thermo Fisher Scientific). Brain sections were incubated for 30 min in blocking solution (0.3% Triton X-100 and 3% bovine serum albumin in phosphate-buffered saline (PBS)). Sections were incubated with primary antibodies in blocking solution overnight at 4 °C and washed three times (10 min each) with 0.05% Triton X-100 in PBS, followed by incubation for 1.5 h at room temperature with secondary antibodies in blocking solution. Sections were washed three times with PBS and DAPI dye (1.0 μg ml$^{-1}$) co-applied during the wash. Sections were mounted using Microscope cover glasses 24 × 60 mm (Marienfeld Superior 0101242) on Menzel-Gläser microscope slides Superfrost Plus (Thermo Fisher Scientific, J1800AMNZ) with ProLong Diamond Antifade Mountant (Invitrogen, P36970).

For S5E2 PV enhancer FingR-PSD95 quantifications, coronal brain slices were cut at 120 μm on a Vibratome (VT1000S, Leica) and cleared with CUBIC-L solution (10% w/v N-butyldiethanolamine, 10% w/v Triton X-100) for 3 h at 37 °C with gentle shaking[59]. Sections were stained with goat anti-parvalbumin antibodies and mounted with Ce3D Tissue Clearing Solution (Biolegend, 427704).

For parvalbumin and WFA analysis, images were acquired on an inverted LSM700 confocal microscope (Zeiss) using 20×/0.45 and 40×/1.30 Apochromat objectives. For quantifications of the cell density of PV interneurons, tile-scan images from the barrel cortex were acquired. For synapse quantifications, images were acquired with a PlanApo 63×/1.4 oil immersion objective.

For primary neocortical neurons in culture, fixation was with 4% PFA in 1× PBS for 15 min. followed by ice-cold methanol (10 min at −20 °C). Cells were blocked (5% donkey serum, 0.3% Triton X-100 in PBS) for 1 h at room temperature and primary antibody incubation was performed overnight at 4 °C in a humidified chamber. Secondary antibody incubation was 1 h at room temperature. Imaging was performed on a widefield microscope (Deltavision, Applied Precision) with a 60× objective (NA 1.42, oil).

### Image analysis

Mean intensity analyses for parvalbumin and WFA stainings were performed in ImageJ with a custom-made script in Python. In brief, H-Watershed was applied to segment PV interneurons on the basis of the tdTomato signal on the soma. To detect the WFA signal, the soma

was eroded and dilated in all optical sections. After applying thresholding, parvalbumin and WFA mean intensity values were automatically calculated and displayed as arbitrary units. Integrity analysis of PNNs was done from PV interneurons with a positive WFA signal (>2,000 arbitrary units). Images were post-processed by conservative deconvolution with the Huygens Deconvolution software with the classic maximum likelihood estimation deconvolution algorithm. Quantitative analyses of the number of peaks and the distance between the peaks were performed by using plot profile function in ImageJ as described[60].

For BRX-reporter experiments, cell identity and reporter intensity were quantified with ImageJ. A region of interest was drawn around the nuclei (marked by DAPI) and the mean intensity was measured for the nuclear GFP signal and normalized to background fluorescence in the same image. Cells were identified on the basis of immunostaining for markers: mCherry (genetically restricted to PV interneurons), NeuN (marking neurons with high intensity in pyramidal cells) and GAD67 (marking all GABAergic cells).

For synapse quantification, images were post-processed by conservative deconvolution with the Huygens Deconvolution software with the classic maximum likelihood estimation deconvolution algorithm. Quantitative analysis was performed using Imaris 9.9.1 by application of spots and surface detection tool.

All data collection and image analysis were done blinded to the genotype or treatment of the mouse. Statistical analyses were done with GraphPad Prism v.9. Images were assembled using ImageJ and Adobe Illustrator software.

### ChIP–seq analysis
For ChIP–seq analysis with cultured neurons, $24 \times 10^6$ cells (DIV14) were cross-linked with 1% formaldehyde for 10 min at room temperature. Cross-linking was stopped by the addition of glycine solution (Cell Signaling Technology, 7005) for 5 min at room temperature. Cells were scraped, pelleted and lysed for 10 min on ice in 100 mM HEPES-NaOH pH 7.5, 280 mM NaCl, 2 mM EDTA, 2 mM EGTA, 0.5% Triton X-100, 1% NP-40 and 20% glycerol. Nuclei were pelleted by centrifugation, washed in 10 mM Tris-HCl pH 8.0, 200 mM NaCl and suspended in 10 mM Tris-HCl pH 8.0, 100 mM NaCl, 1 mM EDTA, 0.5 mM EGTA, 0.1% Na-Deoxycholate and 0.5% N-lauroylsarcosine. Chromatin was sheared using a Covaris Sonicator for 20 min in sonication buffer (SimpleChIP Plus Sonication Kit, Cell Signaling Technology, 57976) to obtain fragments in the range of 200–500 bp. After sonication, sheared chromatin was centrifuged at 16,000g for 20 min at 4 °C and dissolved in 1× ChIP buffer (Cell Signaling Technology, 57976). Input (2%) was taken and the chromatin was incubated with antibodies overnight at 4 °C. Incubation with Protein G magnetic beads, de-cross-linking and elution were performed as described in the SimpleChIP Plus Sonication Kit.

Libraries were generated using the KAPA Hyper Prep (Roche KK8504) according to the manufacturer's instructions, and were amplified by PCR. Library quality was assessed using the High Sensitivity NGS Fragment Analysis Kit (Advanced Analytical DNF-474) on the Fragment Analyzer (Advanced Analytical). Libraries were sequenced paired-end 41 bases on NextSeq 500 (Illumina) using two NextSeq 500 High Output Kit 75-cycles (Illumina, FC-404-1005) loaded at 2.5 pM and including 1% PhiX. Primary data analysis was performed with Illumina RTA v.2.4.11 and Basecalling v.bcl2fastq-2.20.0.422. Two NextSeq runs were performed to compile enough reads (on average per sample in total: 50 million ± 2 million pass-filter reads).

ChIP–seq analysis from P35–P42 mouse cortex was performed using the SimpleChIP Enzymatic Chromatin IP Kit (Cell Signaling Technology, 9003), following the manufacturer's instructions with slight modifications. In brief, neocortices from both hemispheres were cross-linked in 1.5% formaldehyde for 20 min at room temperature. Cross-linking was stopped by the addition of glycine solution for 5 min at room temperature. Tissue was pelleted, washed and disaggregated by using a Dounce homogenizer in 1× PBS containing protease inhibitor cocktail. Nuclei

were pelleted by centrifugation and chromatin was digested by using micrococcal nuclease for 20 min at 37 °C by frequent mixing to obtain fragments in the range of 150–900 bp. Nuclei were pelleted, resuspended in 1× ChIP buffer, sonicated with Bioruptor Pico (Diagenode B01060010) to release sheared chromatin and centrifuged at 9,400g for 10 min at 4 °C. Input (2%) was taken and the chromatin was incubated with primary antibodies overnight at 4 °C. After subsequent incubation with 30 μl Protein G magnetic beads for 2 h at 4 °C, beads were washed three times with low salt, one time with high salt, one time with NP-40 buffer (8 mM Tris-HCl pH 8.0, 2 mM LiCl, 0.8 mM EDTA, 0.4% NP-40 and 0.4% sodium-deoxycholate) and one time with TE buffer (10 mM Tris-HCl, pH 8.0 and 1 mM EDTA) at 4 °C. De-cross-linking and elution were performed as described in the Enzymatic Chromatin IP Kit. Libraries were generated using the NEBNext Ultra II DNA Library Prep Kit for Illumina (New England Biolabs, E7645L) according to the manufacturer's instructions and were amplified by PCR. Library quality was assessed using the High Sensitivity NGS Fragment Analysis Kit (Advanced Analytical, DNF-474) on the Fragment Analyzer (Advanced Analytical) and cleaned up by using 1.0× Vol SPRI beads (Beckman Coulter). Libraries were sequenced paired-end 41 bases on NextSeq 500 (Illumina) using two NextSeq 500 High Output Kit 75-cycles (Illumina, FC-404-1005). Two NextSeq runs were performed to compile enough reads (19–32 million pass-filter reads).

### RNA library preparation and sequencing
Libraries of BMP2-stimulated naïve cortical cultures were prepared from 200 ng total RNA by using the TruSeq Stranded mRNA Library Kit (20020595, Illumina) and the TruSeq RNA UD Indexes (20022371, Illumina). Fifteen cycles of PCR were performed.

Quality checking was performed by using the Standard Sensitivity NGS Fragment Analysis Kit (DNF-473, Advanced Analytical) on the Fragment Analyzer (Advanced Analytical) and quantified (average concentration was 213 ± 15 nmol l$^{-1}$ and average library size was 357 ± 8 bp) to prepare a pool of libraries with equal molarity. The pool was quantified by fluorometry using using the QuantiFluor ONE dsDNA System (E4871, Promega) on a Quantus instrument (Promega). Libraries were sequenced single-reads 76 bases (in addition: 8 bases for index 1 and 8 bases for index 2) on NextSeq 500 (Illumina) using the NextSeq 500 High Output Kit 75-cycles (Illumina, FC-404-1005). Flow lanes were loaded at 1.4 pM of pool and including 1% PhiX. Primary data analysis was performed with Illumina RTA v.2.4.11 and Basecalling v.bcl2fastq-2.20.0.422. The NextSeq runs were performed to compile, on average per sample, 56 million ± 3 million pass-filter reads (illumina PF reads).

For the libraries from control and *Smad1* mutant primary cortical cultures (four biological replicates), 100 ng total RNA was used and library preparation and quality check were performed as described above. Quantification yielded an average concentration of 213 ± 15 nmol l$^{-1}$ and an average library size of 357 ± 8 bp. Libraries were sequenced paired-end 51 bases (in addition: 8 bases for index 1 and 8 bases for index 2) set-up using the NovaSeq 6000 instrument (Illumina). SP Flow-Cell was loaded at a final concentration in flow lanes of 400 pM and including 1% PhiX. Primary data analysis was performed as described above and 43 million ± 5 million per sample (on average) pass-filter reads were collected on 1 SP Flow-Cell.

### ChIP–seq and RNA-seq data analysis
ChIP–seq reads were aligned to the December 2011 (mm10) mouse genome assembly from UCSC[61]. Alignments were performed in R using the qAlign function from the QuasR package1 (v.1.14.0) with default settings[62]. This calls the Bowtie aligner with the parameters "-m 1 –best –strata", which reports only reads that map to a unique position in the genome. The reference genome package (BSgenome.Mmusculus. UCSC.mm10) was downloaded from Bioconductor (https://www.bio-conductor.org). BigWig files were created using qExportWig from the

QuasR package with the bin size set to 50. Peaks were called for each ChIP replicate against a matched input using the MACS2 callpeak function with the default options. Peaks were then annotated to the closest gene and to a genomic feature (promoter, 3′-UTR, exon, intron, 5′-UTR or distal intergenic) using the ChIPseeker R package. The promoter region was defined as −3 kb to +3 kb around the annotated transcription start site. Transcripts were extracted from the TxDb.Mmusculus.UCSC.mm10.ensGene annotation R package. All analyses in R were run in RStudio v.1.1.447 running R v.3.5.1. The enrichment of BMP2-induced peaks over constitutive peaks was analysed by using default settings in the voom−limma analysis software packages[63]. Motif enrichment analysis for BMP2-responsive peaks and constitutive peaks was performed separately by screening for the enrichment of known motifs with the default settings of HOMER[64]. Output motif results with the lowest *P* value and highest enrichment in targets compared to the background sequences were shown for each peak set.

RNA-seq reads were aligned to mm10 using STAR and visualized in the IGV genome browser to determine strand protocol. By using QuasR's qQCReport, read quality scores, GC content, sequence length, adapter content, library complexity and mapping rate were checked and a QC report was generated. Reads with quality scores less than 30, mapping rates lower than 65 or contaminations from noncoding RNAs were not considered for further analysis. For reads that passed QC, QuasR's qCount function was used to count the reads that mapped to annotated exons (from Ensembl genome annotations). Each read was counted once on the basis of its start (if reads are on the plus strand) or end (if reads are on the minus strand) position. For each gene, counts were summed for all annotated exons, without double-counting exons present in multiple transcript isoforms (exon-union model). Correlations between replicates and batch structure were checked by plotting correlation heat maps, PCA plots of samples and scatter plots of normalized read counts. The EdgeR package from R was used to build a model and test for differentially expressed (DE) genes. For DE analysis, counts were normalized using the TMM method (built into edgeR). Any genes with fewer than, in total, 30 reads from all samples were dropped from further analysis. DE analyses were conducted with the voom−limma analysis software packages by using the total number of mapped reads as a scaling factor. Results were extracted from edgeR as tables and used for generating volcano or box plots in ggplot2 in RStudio.

To generate IGV genome browser tracks for ChIP−seq and RNA-seq data, all aligned bam files for each replicate of a given experiment were pooled and converted to BED format with bedtools bamtobed and filtered to be coverted into coverageBED format using bedtools. Finally, bedGraphToBigWig (UCSC-tools) was used to generate the bigWig files displayed on IGV browser tracks in the manuscript.

GO analysis was performed by using the statistical overrepresentation test and cellular component function in PANTHER (http://pantherdb.org/). All genes that were detected as expressed in RNA-seq data were used as reference. GO terms with at least ten genes and at least 1.5-fold enrichment with a false discovery rate of less than 0.05 were considered to be significantly enriched. Significant GO terms were plotted in GraphPad Prism v.9.

## EEG recordings and behavioural monitoring

EEG electrodes were implanted in mice at the age of 12–16 weeks. EEG signals were recorded using two stainless steel screws inserted ipsilaterally into the skull. One was inserted 1.2 mm from the midline and 1.5 mm anterior to bregma, and the other was inserted 1.7 mm from the midline and 2.25 mm posterior from to bregma. Seven days after surgery, mice were transferred to individual behaviour cages with a 12:12 h light−dark cycle and a constant temperature of about 23 °C. Mice had access ad libitum to food and water and were allowed to recover from surgery for seven days. Analysis was performed in individual cages equipped with overhead cameras (FLIR). Mice were connected to an amplifier (A-M Systems 1600) through a commutator. EEG signals were amplified and analog filtered (Gain 500; low-pass filter, 0.3 Hz; high-pass filter, 100 Hz) and then digitized at 200 Hz using Spike2 (CED Micro1401). Spontaneous sleep−wake behaviour was monitored continuously through EEG recordings and video tracking for three weeks. Epileptic episodes were identified at first by inspecting the EEG signals, and were subsequently examined further in the simultaneous video recordings.

## Statistics and reproducibility

All experiments were performed in at least three fully independent replications (on different days, with different mice or cell cultures). Details about the numbers of mice and cultures are provided in the figure legends. When single micrographs or western blots are shown, their results are representative of all independent replicates analysed. Analysis was conducted in R and with GraphPad Prism v.9. Sample sizes were chosen on the basis of previous experiments and literature surveys. No statistical methods were used to predetermine sample sizes. Exclusion criteria used throughout this manuscript were predefined. See the descriptions in the respective sections of the methods. Mice were randomly assigned to treatment groups. Appropriate statistical tests were chosen according to the sample size and the distribution of data points, and are indicated in individual experiments.

## Reporting summary

Further information on research design is available in the Nature Portfolio Reporting Summary linked to this article.

## Data availability

ChIP−seq and RNA-seq data have been deposited at the Gene Expression Omnibus under the accession numbers GSE255466, GSE255562, GSE255563 and GSE25587. DNA plasmids for producing AAV vectors are available through Addgene (including plasmids 20278 and 20279). All other renewable reagents will be distributed by the corresponding author. Source data are provided with this paper.

## Code availability

Custom codes will be provided upon request.

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

**Acknowledgements** We thank G. Aguilar, S. Dixit, W. Yang, Ö. Genç, Z. Chaker, J. Bischofberger, F. Rijli, R. Schneggenburger and K. Tan for support, advice and comments on the manuscript; C. Bornmann, S. Innocenti and E. Perez-Garci for technical assistance; and the Biozentrum

Imaging Core Facility, the Centre for Transgenic Models and the Quantitative Genomics Facility Basel for expert support. This work was financially supported by a Fellowship of Excellence from the Biozentrum, University of Basel to Z.O., an EMBO Long-Term Fellowship to M.P. (ALTF 672-2022), an EMBO Long-Term Fellowship to R.O. (ALTF 378-2020) and grants from the Swiss National Science Foundation to P.S. (grants 179432, 154455 and 209273). M.S. was financially supported by the NCCR RNA & Disease from the Swiss National Science Foundation and the Scheiffele laboratory is an associated member of the NCCR RNA & Disease network.

**Author contributions** All authors contributed to the design and analysis of experiments. Z.O. and N.S. performed genetic and in vivo manipulations. Z.O. and M.P. performed electrophysiological recordings. R.O. and M.S. contributed to analyses of ChIP–seq and RNA-seq data. V.B. performed EEG recordings. Z.O., D.S., N.S. and K.K. performed molecular biology procedures. Z.O. and P.S. wrote the manuscript, and feedback and editing were provided by all authors.

**Funding** Open access funding provided by University of Basel.

**Competing interests** The authors declare no competing interests.

**Additional information**
**Correspondence and requests for materials** should be addressed to Peter Scheiffele.

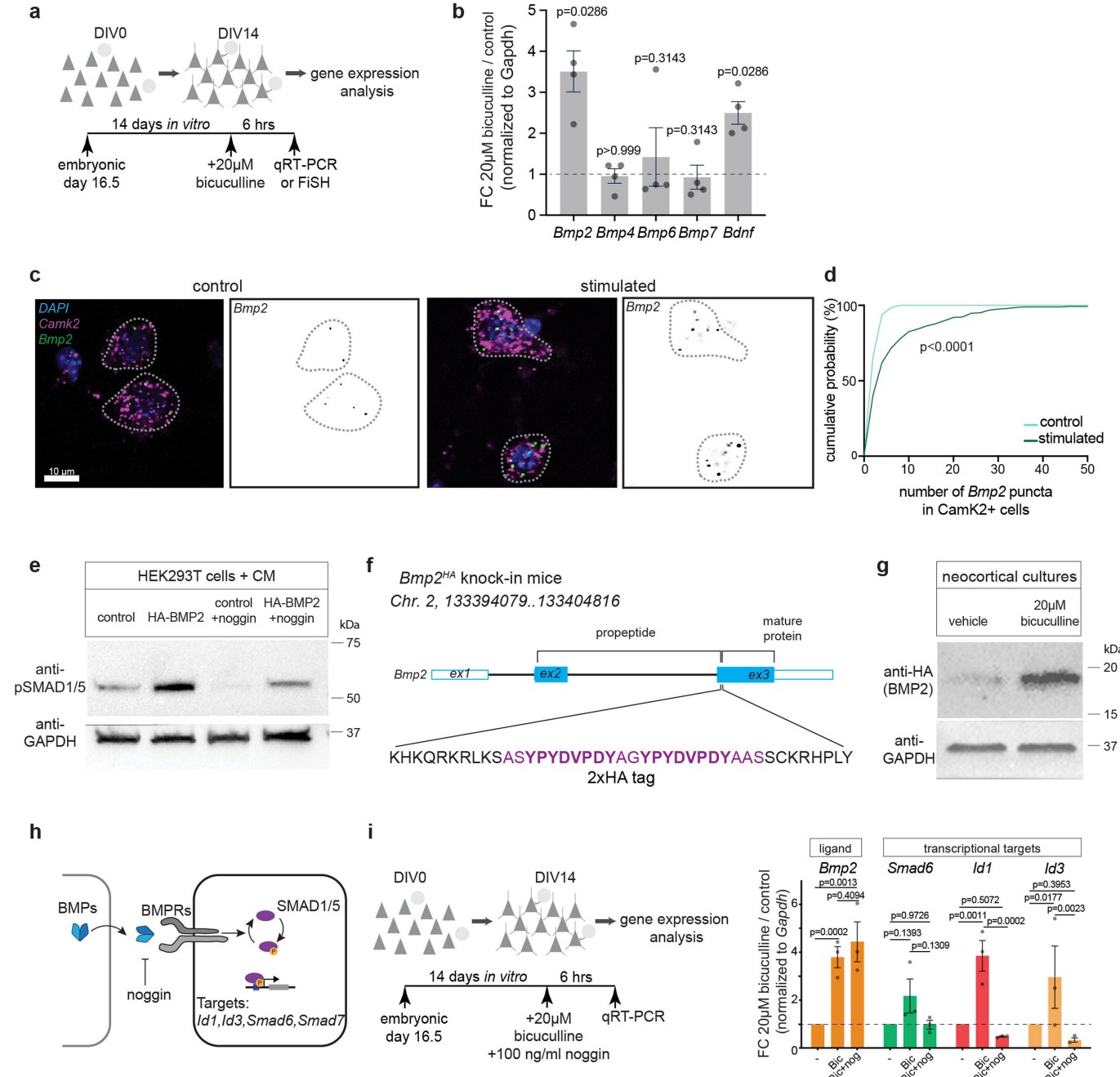

**Extended Data Fig. 1 | Increased neuronal network activity triggers the upregulation of BMP2 in neocortical glutamatergic neurons in vitro.**
**a**, Schematic representation of cortical cultures and pharmacological activity manipulation. **b**, qPCR assessment of *Bmp2*, *Bmp4*, *Bmp6*, Bmp7 and *Bdnf* transcripts in DIV14 neocortical cultures treated with 20 µM bicuculline for 6 h expressed as fold change (FC) compared to untreated cultures. All expression values were normalized to *Gapdh* (N = 4 independent cortical cultures, total of 3 technical replicates, mean and ±SEM, Mann-Whitney test, two-tailed). **c**, *Bmp2* transcript levels by fluorescence in situ hybridization (FiSH) in *Camk2*-positive glutamatergic neurons in naïve and stimulated (with 25 mM KCl) neocortical cultures). **d**, Cumulative distribution of *Bmp2* FiSH signal per cell in control and stimulated neurons (N = 3 independent cortical cultures, n = 189 for control and n = 286 cells for stimulated cortical cultures, Kolmgorov–Smirnov test). **e**, Confirming functional signalling for HA-epitope-tagged BMP2 in cultured cells. Western blot for phosphorylated SMAD1/5 (anti-pSMAD1/5) in cultured human embryonic kidney cells (HEK293T) treated with conditioned medium (CM) from control or HA-BMP2-expressing cells containing or lacking 100 ng/ml noggin. **f**, Illustration of Crispr-based knock-in strategy for introduction of an epitope tag into the endogenous mouse *Bmp2* locus. A double HA tag sequence and flanking homology arms were encoded in a single-stranded DNA oligo and were inserted in *Bmp2* exon 3 (ex3) at a Crispr/Cas9 cleavage site through homology-directed repair. The 2x HA tag is positioned at the N terminus of the mature BMP2 protein. Resulting homozygous *Bmp2HA/HA* knock-in mice were viable and fertile. **g**, Western blot with anti-HA antibodies of lysate from cultured neocortical neurons from *Bmp2HA/HA* knock-in mice (DIV14) either naïve or treated for 24 h with 20 µM bicuculline (N = 3 independent cortical cultures). BMP2HA expression levels in vivo could not be reliably assessed, likely due to its low abundance in the complex tissue samples. **h**, Illustration of inhibition of BMP signalling by the extracellular antagonist noggin. **i**, qPCR assessment of *Bmp2*, *Smad6*, *Id1* and *Id3* transcripts expressed as fold change in bicuculline (bic, 20 µM for 6 h) and Bic+Nog (20 µM bicuculline and 100 ng/ml noggin for 6 h) compared to naïve cultures (N = 3 independent cortical cultures, each with 3 technical replicates, mean and ±SEM, two-way ANOVA with Tukey's post-hoc test).

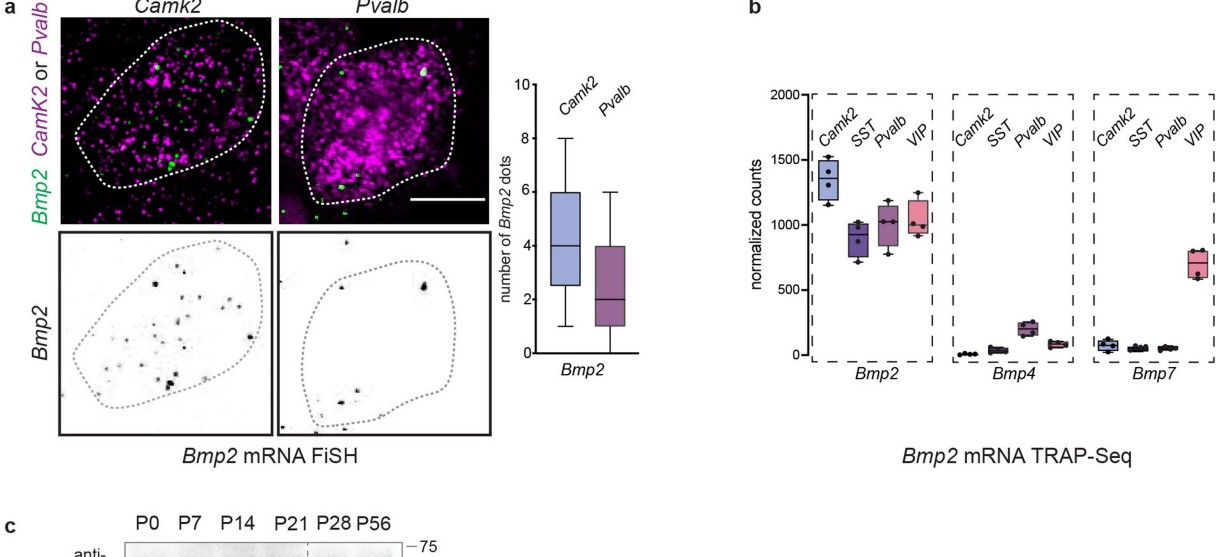

**Extended Data Fig. 2 | Expression of BMP signalling components in the adult mouse neocortex. a**, Quantification of *Bmp2* mRNA expression in *Camk2*⁺ and *Pvalb*⁺ neurons in layer 2/3 of mouse barrel cortex (P25-30) assessed by FiSH (N = 3 mice, n = 57 cells/*Camk2*+ and n = 45 cells/*Pvalb*+). Horizontal lines mark median, lower and upper bounds of boxes represent 25th to 75th percentiles, whiskers indicate 10–90 percentile. **b**, mRNA expression of *Bmp2*, *Bmp4*, *Bmp7* in P25 mouse neocortex in genetically defined *Camk2*⁺ principal neurons and

somatostatin⁺ (SST), PV, and Vasoactive intestinal peptide⁺ (VIP) interneurons extracted from SPLICECODE database of TRAP-Seq analysis[51]. N = 4 mice/group, horizontal lines mark median, lower and upper bounds of boxes represent 25th to 75th percentiles and whiskers indicate min to max. **c**, Developmental expression levels assessed by western blot of BMP receptor type 2 (BMPR2), transcriptional mediators (SMAD1 and SMAD5) and their active complex (pSMAD1/5/9) in the mouse neocortex (P0 to P56).

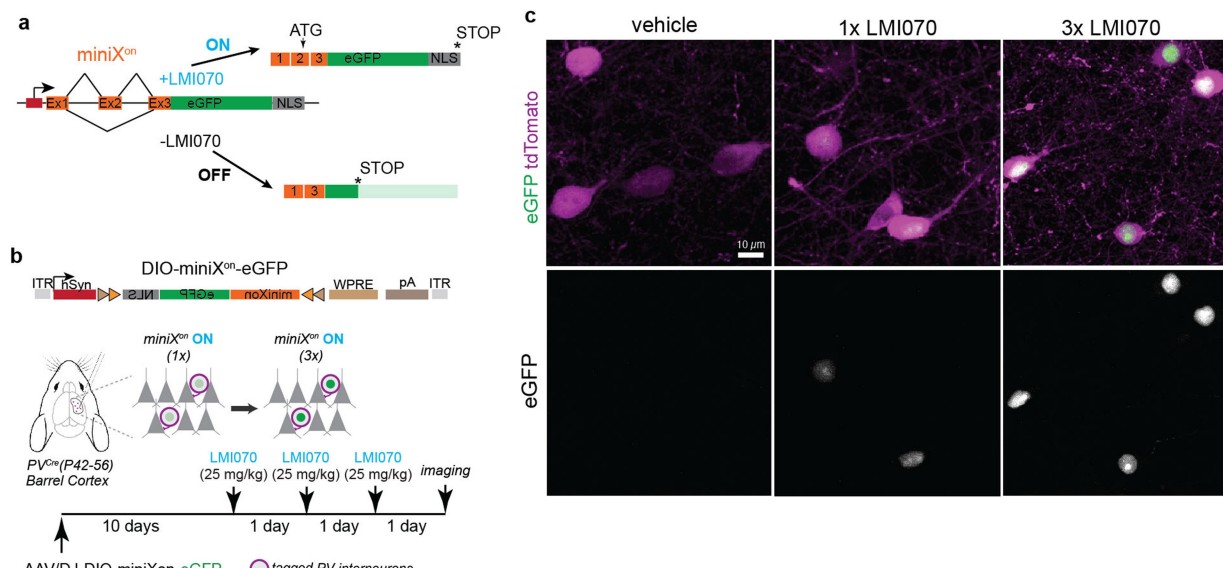

**Extended Data Fig. 3 | Regulated expression from chemically gated AAV X^on in barrel-cortex PV interneurons. a**, Schematic illustration of miniX^on regulation of protein expression[33]. In the presence of the small molecule LMI070, alternative splicing of the cassette shifts to include a translational start codon in exon 2 (Ex2) and, thus, turns on expression of nuclear targeted eGFP reporter protein (NLS-eGFP). In the absence of LMI070, the AUG start codon-containing exon is skipped and translation does not occur in the correct reading frame. **b**, Schematic diagram for cre-dependent expression of miniX^on constructs in PV interneurons by AAV injection into the barrel cortex of adult *PV^cre* mice. **c**, Representative images for nuclear NLS-eGFP expression in PV interneurons of mice treated by oral gavage with vehicle or 25 mg/kg LMI070 (1x or 3x in 24-hour intervals).

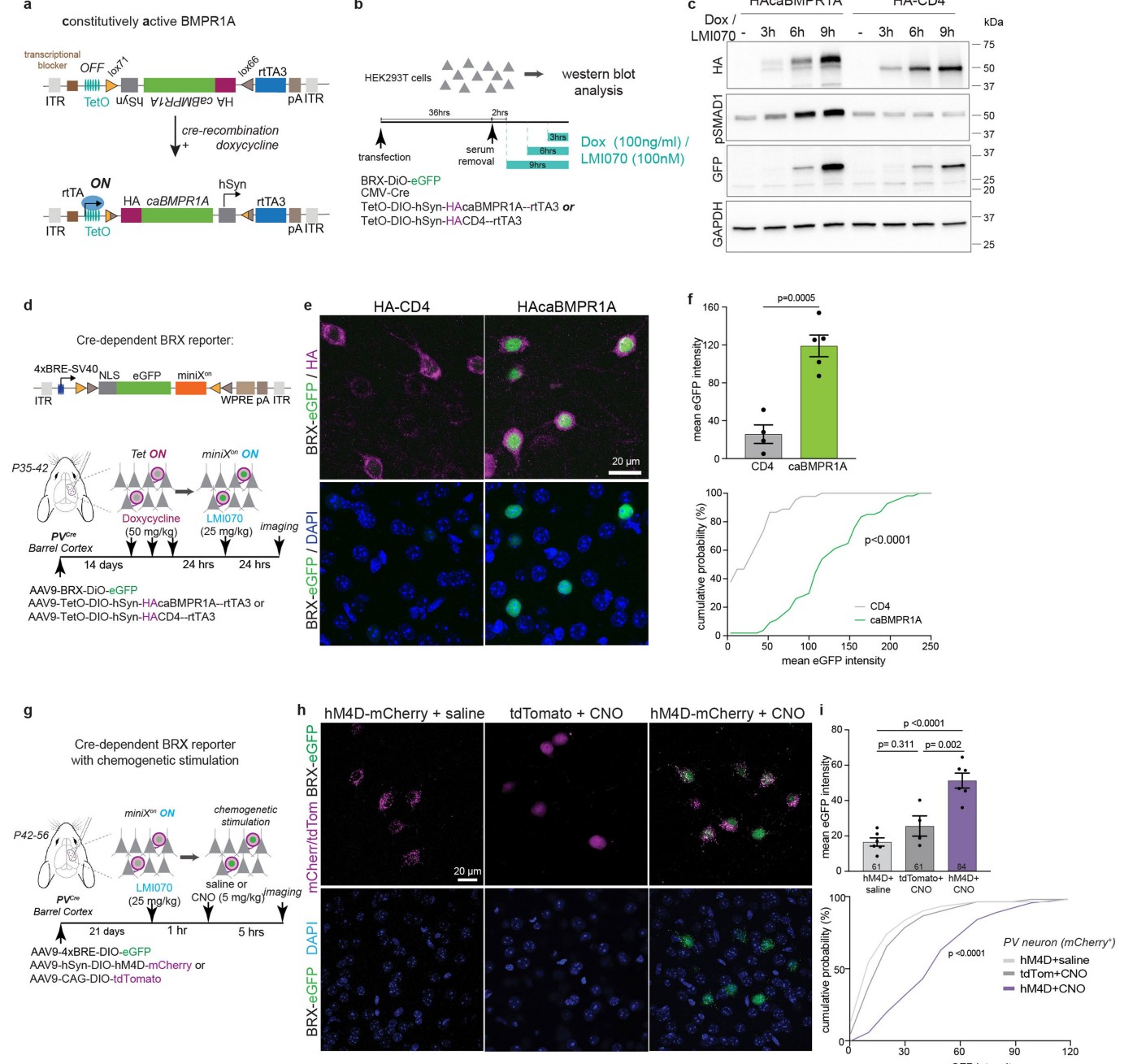

**Extended Data Fig. 4 | Cre- and doxycycline-dependent expression vector for the constitutively active BMPR1A receptor and BRX reporters. a**, The vector contains a cre-dependent (lox71-lox66) inversion cassette encoding the open reading frame of the HA-epitope-tagged receptor (HA-caBMPR1A) and human synapsin promoter. After inversion, the reverse transcriptional activator rtTA3 is expressed from the synapsin promoter. In presence of doxycycline, rtTA3 drives transcription of HA-caBMPR1A from Tet operator sequences (TetO). **b**, Experimental paradigm for testing regulated expression of HA-caBMPR1A and activation of the BRX-eGFP reporter in vitro. **c**, Western blot analysis of cell lysates from transiently transfected HEK293 cells expressing BRX-DiO-eGFP, cre recombinase under control of CMV promoter, and HA-caBMPR1A or a HA-CD4 control protein under control of the TetO elements. After chemical induction with doxycycline and LMI070, expression of HA-caBMPR1A results in an elevation of pSMAD1 signal and accumulation of eGFP from the BRX reporter. The low level of eGFP expression seen in the HA-CD4 control condition represents a low level of leakiness of expression from 4xBRE elements or activation due to endogenous BMP signalling events. **d**, Vector for cre-dependent BRX reporter where the BRX-NLS-eGFP cassette is inverted and flanked by loxP sites in a DiO configuration. Schematic diagram for

cre-dependent co-expression of BRX-DiO-eGFP and HA-caBMPR1A constructs in PV interneurons by AAV injection into the barrel cortex of adult $PV^{cre}$ mice. **e**, BRX reporter signal in barrel cortex layer 2/3 of $PV^{Cre}$ mice. **f**, Bar graph for mean ± SEM of nuclear eGFP intensity per mouse (N = 4-5 mice/group, n = 45–54 cells per condition, unpaired t-test, two-tailed) and cumulative distribution of eGFP reporter intensity per PV interneuron (n = 45–54 cells per condition, Kolmogorov–Smirnov test). **g**, Cre-dependent co-expression of BRX-DiO-eGFP and DiO-hM4D-mCherry or DiO tdTomato negative control constructs in PV interneurons by AAV injection into the barrel cortex of P42–P56 $PV^{cre}$ mice. **h**, Cre-dependent BRX reporter in barrel cortex layer 2/3 for negative control conditions (saline injection in mice expressing hM4D-mCherry or CNO injection in mice expressing tdTomato) or chemogenetic stimulation (CNO injection in mice with hM4D expression). **i**, Bar graph for mean ± SEM of nuclear eGFP intensity in PV cells identified by mCherry or tdTomato expression (N = 4–6 mice/group, n = 61–84 cells per condition, one-way Anova with Tukey's multiple comparisons multiple comparisons) and cumulative distribution of eGFP reporter intensity per PV interneuron (Kolmogorov–Smirnov test). Scale bars in **e** and **h** are 20 μm.

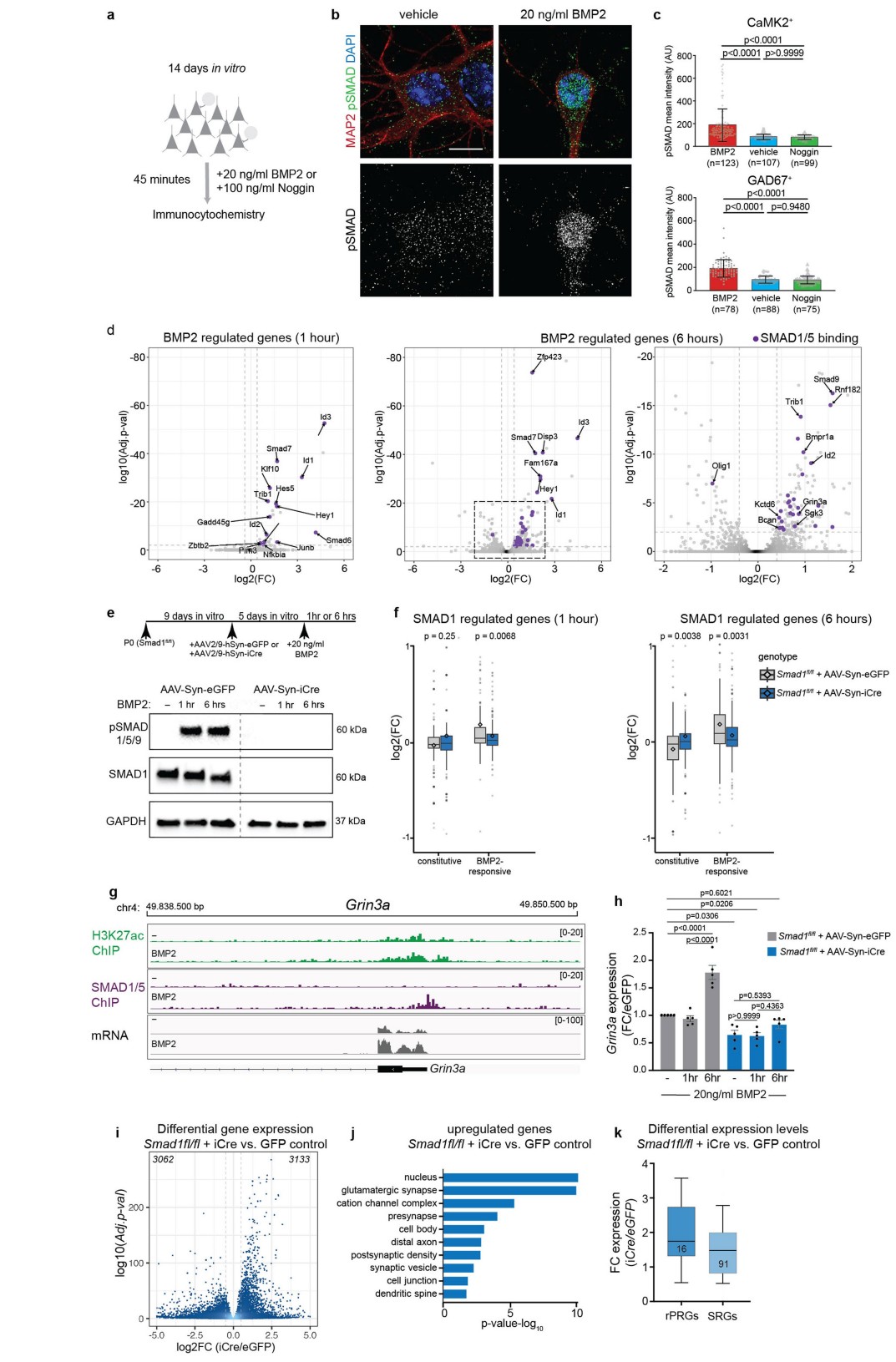

**Extended Data Fig. 5** | See next page for caption.

**Extended Data Fig. 5 | BMP–SMAD1 signalling and target genes in neocortical neurons in vitro. a**, Schematic representation of cortical cultures and BMP pathway manipulations. **b**, Immunostaining of naïve or BMP2-stimulated (20 ng/ml for 45 min) cultured neocortical neurons (DIV14) with antibodies to the neuronal marker microtubule associated protein-2 (MAP2) and pSMAD1/5/9 (activated SMAD). **c**, Quantification of nuclear pSMAD intensity in cultured CaMK2$^+$ glutamatergic neurons and GAD67$^+$ GABAergic neurons from BMP2-treated (20 ng/ml 45 min), vehicle-treated, and noggin-treated (100 ng/ml, 45 min) cortical cultures (N = 3 independent cultures, number of cells are indicated on the plot, one-way Anova followed by Tukey's multiple comparisons test, the bar graphs show then means ± SEM). **d**, Volcano plot of RNA-Seq expression data from neocortical cultures (DIV14) stimulated with BMP2 (20 ng/ml) for one hour (left) or 6 h (middle and right). Log2 fold change (FC) of expression values for stimulated over non-stimulated cells and log10 adjusted p-values are displayed (adj. p-value with correction for multiple comparisons). Direct SMAD1/5 targets identified in ChIP–seq that are significantly regulated are marked in purple. Grey dashed lines indicate cut-offs to consider genes significantly regulated (30% change and adj. p-value of 0.01). The right panel shows an enlargement of the area indicated by the black dashed box (in middle panel) to better visualize genes the moderate but significantly regulation. **e**, Experimental design and western blot for detection of SMAD1 and pSMAD1/5/9 protein levels in control (AAV-Syn-eGFP) and neuron-specific *Smad1* conditional knockout (AAV-Syn-iCre) cultured neocortical neurons (DIV14), either naïve (-) or treated with recombinant BMP2 (20 ng/ml) for 1 hr or 6 hrs (representative of N = 3 independent cortical cultures). **f**, Differential gene expression in neocortical cultures (DIV14) from *Smad1$^{fl/fl}$* mice infected with AAV-Synapsin promoter-eGFP (negative control, grey) or AAV-Synapsin promoter-iCre viruses (resulting in neuron-specific knockout, blue). Box plots show the log2 fold change in gene expression 1 hr (left) or 6 hrs (right) after BMP2 stimulation as compared to non-stimulated cultures. Genes with constitutive and BMP2-responsive SMAD1/5-binding events as identified by ChIP–seq are plotted separately. The statistically significant increase in expression of genes with constitutive SMAD1/5-binding events suggests that these genes are normally repressed by SMAD1. Horizontal lines mark the median, diamonds mark the mean, lower and upper bounds of boxes represent 25th to 75th percentiles, upper whisker indicates the largest observation less than or equal to upper bound + 1.5 * IQR where IQR is the inter-quartile range, or distance between the first and third quartiles, lower whisker indicates the smallest observation greater than or equal to lower hinger - 1.5 * IQR (N = 4 cultures/condition, Wilcoxon test). **g**, Example of IGV genome browser ChIP–seq tracks displaying H3K27ac (green), SMAD1/5 (purple) and RNA-seq signal for SMAD1/5 targets *Grin3a* in naïve (-) and BMP2-stimulated cultures. **h**, qPCR analysis of *Grin3a* mRNA abundance in AAV-Syn-eGFP infected *versus* AAV-Syn-iCre infected *Smad1$^{fl/fl}$* neocortical cultured neurons. Fold change (FC) relative to unstimulated cells is shown for 1 h and 6 h stimulation with 20 ng/ml BMP2. Bar graphs show means ± SEM (N = 5 per condition, one-way ANOVA with Tukey's multiple comparisons). **i**, Volcano plot of differential gene expression in naïve (i.e. not BMP-stimulated) *Smad1$^{fl/fl}$* cortical cultures infected with AAV-Syn-iCre *versus* AAV-Syn-eGFP (adj. p-value with correction for multiple comparisons). Dashed lines indicate log$_2$FC:0.4 and -log10Adj.-p-val: 2 chosen as thresholds for significant regulation. Number of significantly down- and upregulated genes are indicated on the top. **j**, Top ten enriched cellular component GO terms for genes upregulated in conditional *Smad1* mutant cells (*Smad1$^{fl/fl}$* infected with AAV-Syn-iCre) in unstimulated cortical cultures (Fisher's exact test with Bonferroni correction for multiple testing). **k**, Expression levels of neuronal-activity-regulated rapid Primary Response Genes (rPRGs) and Secondary Response Genes (SRGs) as defined in[65] in conditional *Smad1* mutant cells (*Smad1$^{fl/fl}$* infected with AAV-Syn-iCre) compared to control AAV-Syn-eGFP infected cultures. All 16 rPRGs and 91 SRGs reliably detected in our dataset were plotted. Horizontal lines mark the median, lower and upper bounds of boxes represent 25th to 75th percentiles, and whiskers indicate 10–90 percentile.

**a**

Xph20-eGFP

5 µm

FingR PSD95-eGFP

FingR PSD95-mGreenL

FingR PSD95-mGreenL
vGlut1

**b**  Quantification of dendritic glutamatergic synapses onto PV interneurons

**1**

● FingR
● vGlut1
PV tdT+

→ tdTomato positive PV cell with postsynaptic binder, presynaptic antibody marker and nuclear stain (DAPI)

**2**

~ 30 µm

→ Filament detection with tdTomato channel masks dendrite and allows measurement of selected stretch

**3**

→ Spots are masked in mGreenLantern channel and spots in distance closer to 1 µm in distance are quantified

**c**  Quantification of GABAergic PV-PV synapses formed onto the soma of PV interneurons

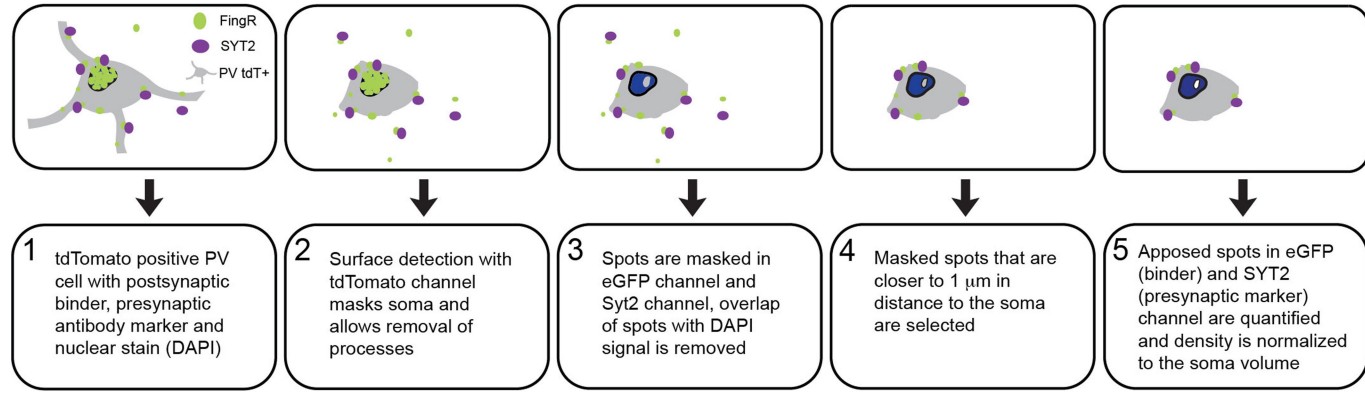

● FingR
● SYT2
PV tdT+

**1** tdTomato positive PV cell with postsynaptic binder, presynaptic antibody marker and nuclear stain (DAPI)

**2** Surface detection with tdTomato channel masks soma and allows removal of processes

**3** Spots are masked in eGFP channel and Syt2 channel, overlap of spots with DAPI signal is removed

**4** Masked spots that are closer to 1 µm in distance to the soma are selected

**5** Apposed spots in eGFP (binder) and SYT2 (presynaptic marker) channel are quantified and density is normalized to the soma volume

**Extended Data Fig. 6 | Optimization of intrabody labelling and quantification of synaptic innervation of PV interneurons.** We optimized the original FingR-PSD95-eGFP constructs[39] by introducing the neuron-optimized fluorophore mGreenLantern[66], and placing the cDNA under control of the neuron-specific synapsin promoter. We then compared FingR-PSD95-mGreenLantern with FingR-PSD95-eGFP and a PSD95 paralog-specific Xph20-EGFP intrabody[67] by stereotaxic injection of cre-dependent AAVs into the barrel cortex of adult (P56–P72) $PV^{cre}$ mice. In these experiments, the FingR-PSD95-mGreenLantern constructs yielded the most reproducible and discrete labelling of glutamatergic postsynaptic sites with little cytoplasmic or non-synaptic labelling. **a**, Representative images of PV interneurons expressing Xph20-eGFP, PSD95FingR-eGFP, PSD95FingR-mGreenLantern. Scale bar in top panel is 5 µm. Co-immunostaining with the glutamatergic presynaptic marker VGlut1 (magenta) reveals extensive overlap with the postsynaptic FingR-PSD95-mGreenLantern marker. **b**, Illustration of 3D quantification protocol for glutamatergic synapses on PV interneuron dendrites with IMARIS software. **c**, Quantification protocol developed in IMARIS to quantify perisomatic GABAergic synapses labelled with FingRGPHN-eGFP intrabodies and co-stained with Syt2 on PV interneurons.

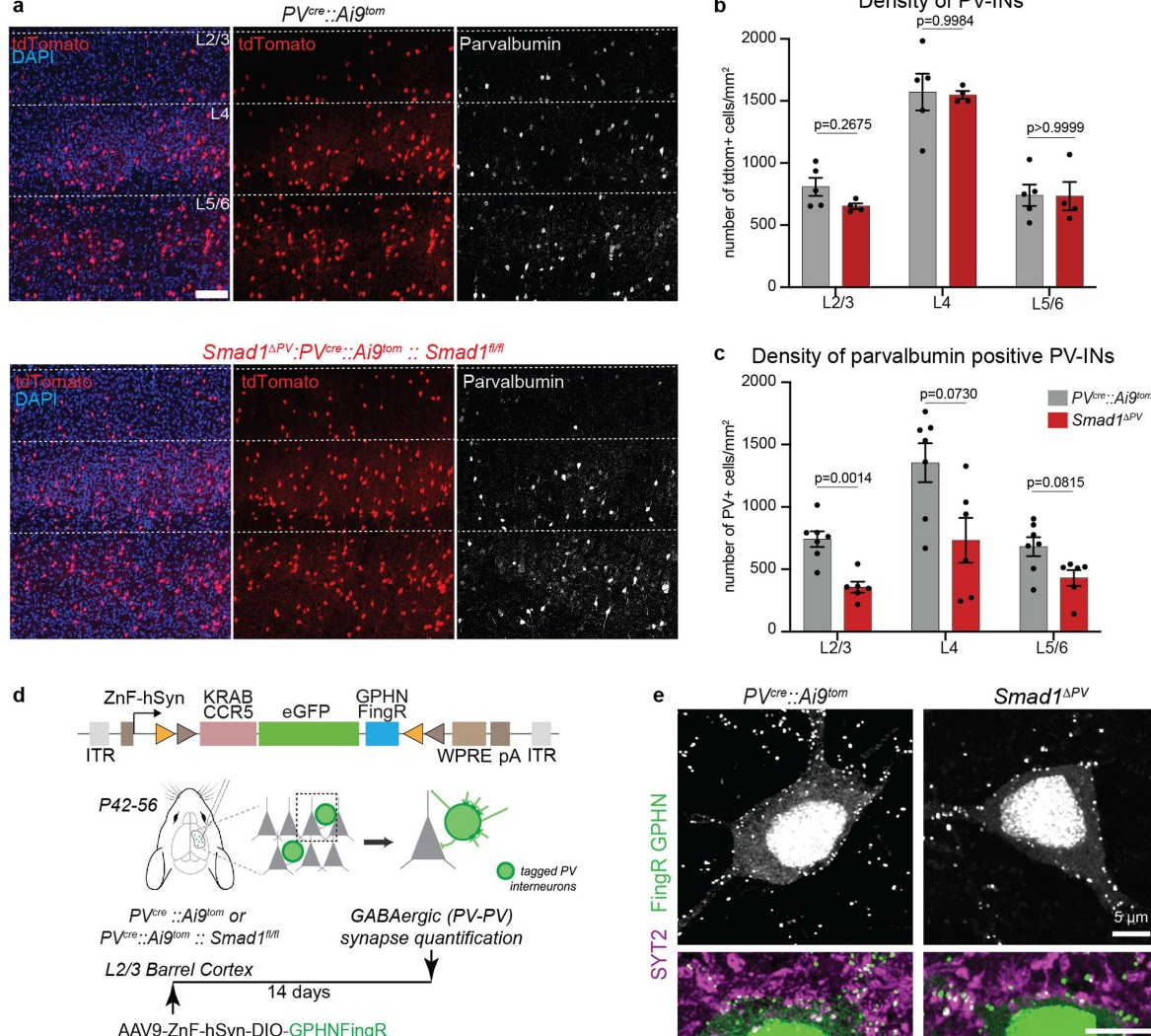

**Extended Data Fig. 7 | Normal PV interneuron density in the barrel cortex of _Smad1^{ΔPV}_ mice. a**, Representative images of coronal sections of adult mouse (P56–P72) barrel cortex of _PV^{cre}::Ai9^{tom}_ mice (left) and _Smad1^{ΔPV}_ mice (right) displaying nuclear DAPI, tdTomato, and anti-parvalbumin immunoreactivity. Scale bar is 100 μm. **b**, Density of genetically labelled tdTomato⁺ PV interneurons in barrel cortex of _PV^{cre}::Ai9^{tom}_ mice and _Smad1^{ΔPV}_ mice (N = 4-5 mice/genotype, n = 2 sections per genotype, mean cell density/mouse and SEM, two-way Anova followed with Sidak's multiple comparisons test). **c**, Quantification of density of parvalbumin immunoreactive PV interneurons across layers in barrel cortex

of _PV^{cre}::Ai9^{tom}_ mice (left) and _Smad1^{ΔPV}_ mice (N = 6-7 mice/genotype, n = 2 sections per genotype, mean cell density/mouse and SEM, two-way ANOVA followed with Sidak's multiple comparisons test). **d**, Schematic representation of AAV-driven, cre-recombinase-dependent intrabody probes for GABAergic synapses (GPHN-FingR), fused to eGFP and a CCR5-KRAB transcriptional repressor for autoregulation of probe expression. Thus, excess probe accumulates in the nucleus. **e**, Synapses formed onto control (_PV^{cre}::Ai9^{tom}_) and _Smad1_ conditional knockout (_Smad1^{ΔPV}_) PV interneurons. Scale bars are 5 μm.

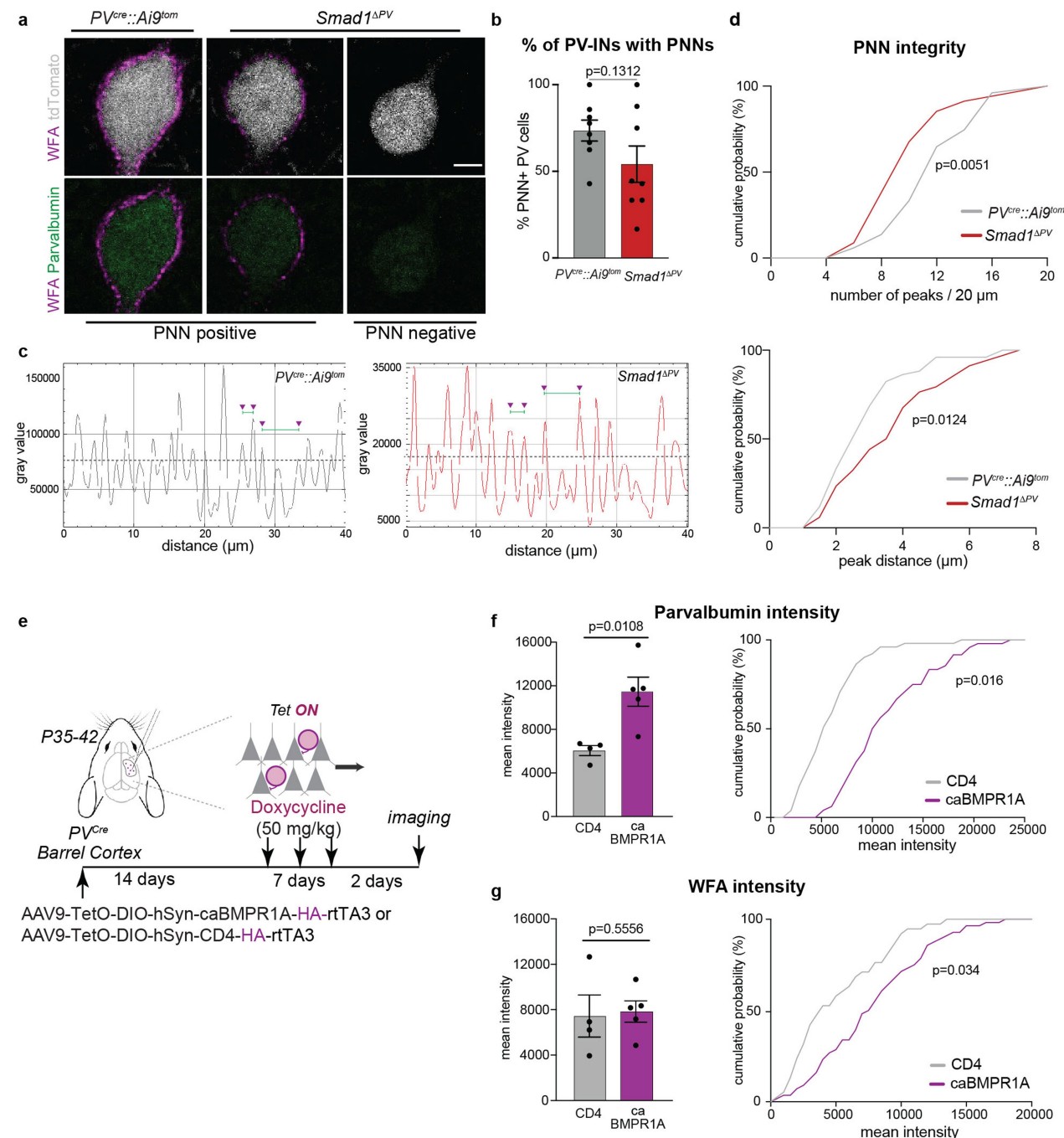

**Extended Data Fig. 8 | Genetic activation of BMP signalling in PV interneurons in the adult somatosensory cortex results in an increase in parvalbumin expression. a**, High-magnification views of individual PV interneurons in $PV^{cre}::Ai9^{tom}$ and $Smad1^{\Delta PV}$ mice, stained with WFA and anti-parvalbumin antibodies. For $Smad1^{\Delta PV}$ mice two examples are given – one PNN-positive and one PNN-negative cell. Scale bar is 5 µm. **b**, Quantification of PNN-positive and -negative cells (8 mice per genotype, mean and SEM, unpaired t-test, two-tailed). **c**, Line graphs used for quantification of PNN integrity in PNN-positive cells from $PV^{cre}::Ai9^{tom}$ and $Smad1^{\Delta PV}$ mice. WFA staining intensity along a 20–40 µm line drawn at the centre of the PNN structure is plotted. The number of peaks above a relative intensity threshold set at 50% of maximum peak height was quantified as well as inter-peak distances. For each plot 4 peaks and 2 examples of inter-peak intervals are marked. **d**, Cumulative frequency plots for peak density and peak distance as described in **c**. 51- and 35-line graphs per genotype, derived from 8 mice per genotype. K.S. test. **e**, Schematic diagram

for expression of constitutively active BMPR1A (caBMPR1A) in PV interneurons. Recombinant AAVs encoding cre- and doxycycline-dependent expression cassettes (see Extended Data Fig. 4 for details) were injected into the barrel cortex of adult $PV^{cre}$ mice. Expression of the caBMPR1A or the HA-CD4 control protein was induced by three intraperitoneal injections of doxycycline administered over one week and cells were analysed 2 days after the final injection. **f**, Mean intensity per mouse and cumulative frequency of parvalbumin staining intensity in genetically identified PV interneurons marked by the HA epitope on caBMPR1A or the CD4 control protein. N = 4-5 mice, n = 48–51 cells. Mean and SEM, unpaired t-test (two-tailed) for bar graphs, and K.S. test for cumulative frequency plots. **g**, Mean intensity per mouse and cumulative frequency of WFA staining intensity surrounding genetically identified PV interneurons marked by the HA epitope on caBMPR1A or the CD4 control protein as in **f**.

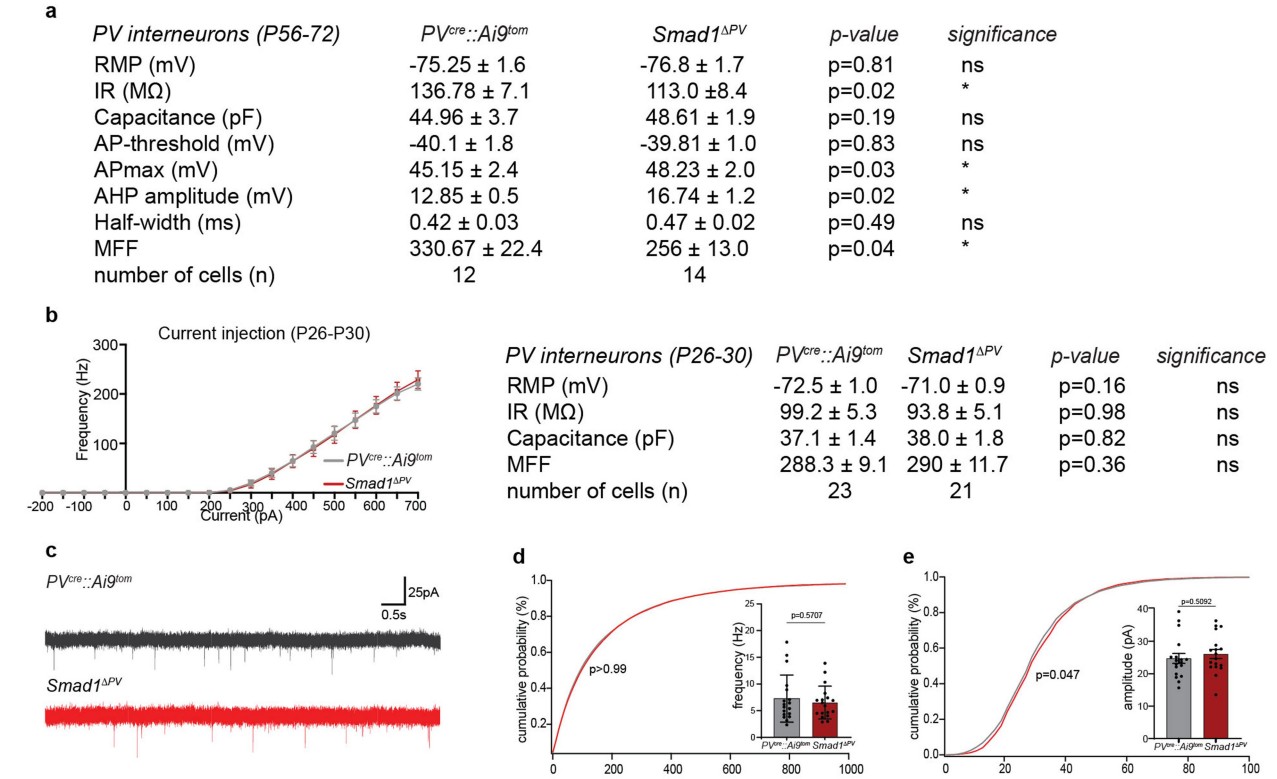

**a**

| PV interneurons (P56-72) | $PV^{cre}::Ai9^{tom}$ | $Smad1^{\Delta PV}$ | p-value | significance |
|---|---|---|---|---|
| RMP (mV) | -75.25 ± 1.6 | -76.8 ± 1.7 | p=0.81 | ns |
| IR (MΩ) | 136.78 ± 7.1 | 113.0 ±8.4 | p=0.02 | * |
| Capacitance (pF) | 44.96 ± 3.7 | 48.61 ± 1.9 | p=0.19 | ns |
| AP-threshold (mV) | -40.1 ± 1.8 | -39.81 ± 1.0 | p=0.83 | ns |
| APmax (mV) | 45.15 ± 2.4 | 48.23 ± 2.0 | p=0.03 | * |
| AHP amplitude (mV) | 12.85 ± 0.5 | 16.74 ± 1.2 | p=0.02 | * |
| Half-width (ms) | 0.42 ± 0.03 | 0.47 ± 0.02 | p=0.49 | ns |
| MFF | 330.67 ± 22.4 | 256 ± 13.0 | p=0.04 | * |
| number of cells (n) | 12 | 14 | | |

**b**

| PV interneurons (P26-30) | $PV^{cre}::Ai9^{tom}$ | $Smad1^{\Delta PV}$ | p-value | significance |
|---|---|---|---|---|
| RMP (mV) | -72.5 ± 1.0 | -71.0 ± 0.9 | p=0.16 | ns |
| IR (MΩ) | 99.2 ± 5.3 | 93.8 ± 5.1 | p=0.98 | ns |
| Capacitance (pF) | 37.1 ± 1.4 | 38.0 ± 1.8 | p=0.82 | ns |
| MFF | 288.3 ± 9.1 | 290 ± 11.7 | p=0.36 | ns |
| number of cells (n) | 23 | 21 | | |

**Extended Data Fig. 9 | Functional properties of *Smad1*-deficient PV interneurons. a**, Intrinsic and action potential properties of layer 2/3 PV interneurons from P56-72 *PV^cre^::Ai9^tom^* mice and *Smad1^ΔPV^* mice. RMP: resting membrane potential, IR: input resistance, MFF: maximum firing frequency, AP: action potential, AHP: afterhyperpolarization (N = 4 mice, n = 12 cells for *PV^cre^::Ai9^tom^* and N = 4, n = 14 cells for *Smad1^ΔPV^*, mean and SEM Kolmogorov–Smirnov test). **b**, Comparison of firing frequencies of PV interneurons at given currents and their intrinsic properties from P26-30 *PV^cre^::Ai9^tom^* mice (grey) and *Smad1^ΔPV^* mice (red). RMP: resting membrane potential, IR: input resistance, MFF: maximum firing frequency (N = 4 mice, n = 23 cells for PVcre::Ai9tom

and N = 4 mice, n = 21 cells for *Smad1^ΔPV^* mice, mean and SEM, Kolmogorov–Smirnov test). **c**, Representative traces of mEPSC recordings from control (grey) and *Smad1^ΔPV^* (red) PV interneurons in acute slice preparations from adolescent mice (P26–P30). **d**, Frequency distribution of interevent intervals (Kolmogorov–Smirnov test) and mean mEPSC frequency (mean ± SEM for n = 18 cells/genotype, from N = 3 mice. Kolmogorov–Smirnov test). **e**, Frequency distribution of mEPSC amplitudes (Kolmogorov–Smirnov test) and mean mEPSC amplitude (mean ± SEM for n = 18 cells/genotype, from N = 3 mice. Kolmogorov–Smirnov test).

# Reporting Summary

## Statistics

For all statistical analyses, confirm that the following items are present in the figure legend, table legend, main text, or Methods section.

| n/a | Confirmed | |
|---|---|---|
| ☐ | ☒ | The exact sample size (*n*) for each experimental group/condition, given as a discrete number and unit of measurement |
| ☐ | ☒ | A statement on whether measurements were taken from distinct samples or whether the same sample was measured repeatedly |
| ☐ | ☒ | The statistical test(s) used AND whether they are one- or two-sided <br> *Only common tests should be described solely by name; describe more complex techniques in the Methods section.* |
| ☐ | ☒ | A description of all covariates tested |
| ☐ | ☒ | A description of any assumptions or corrections, such as tests of normality and adjustment for multiple comparisons |
| ☐ | ☒ | A full description of the statistical parameters including central tendency (e.g. means) or other basic estimates (e.g. regression coefficient) AND variation (e.g. standard deviation) or associated estimates of uncertainty (e.g. confidence intervals) |
| ☒ | ☐ | For null hypothesis testing, the test statistic (e.g. *F*, *t*, *r*) with confidence intervals, effect sizes, degrees of freedom and *P* value noted <br> *Give P values as exact values whenever suitable.* |
| ☒ | ☐ | For Bayesian analysis, information on the choice of priors and Markov chain Monte Carlo settings |
| ☒ | ☐ | For hierarchical and complex designs, identification of the appropriate level for tests and full reporting of outcomes |
| ☒ | ☐ | Estimates of effect sizes (e.g. Cohen's *d*, Pearson's *r*), indicating how they were calculated |

*Our web collection on statistics for biologists contains articles on many of the points above.*

## Software and code

Policy information about availability of computer code

| Data collection | pClamp 11 (Molecular devices) <br> FLIR (Teledyne) <br> Zen (Zeiss) <br> SoftWorx 4.1.2 (API DeltaVision Core) <br> NextSeq 500 (Illumina) <br> NovaSeq 6000 (Illumina) |
|---|---|

| Data analysis | Igor Pro8 (WaveMetrics)<br>Neuromatic v3.0<br>Clampfit 10 (Molecular Devices)<br>Panther16.0 (pantherdb.org)<br>Hyugens Professional (Scientific Volume Imaging)<br>IMARIS 9.9.1 (Oxford Instruments)<br>ImageJ (NHI, 2.9.0/1.53t)<br>R Studio (Posit Software, PBC), Version 2022.12.0+353 (2022.12.0+353)<br>MACS2 (CZI EOSS) v2.1.3.3<br>limma/voom (Bioconductor.org), Limma. version 3.58.1<br>Homer (http://homer.ucsd.edu/homer/motif)<br>PANTHER 16 (http://pantherdb.org/)<br>Prism 9 (GraphPad )<br>ANY-MAZE v5.23 (Stoelting) |
|---|---|

For manuscripts utilizing custom algorithms or software that are central to the research but not yet described in published literature, software must be made available to editors and reviewers. We strongly encourage code deposition in a community repository (e.g. GitHub). See the Nature Portfolio guidelines for submitting code & software for further information.

## Data

Policy information about availability of data

All manuscripts must include a data availability statement. This statement should provide the following information, where applicable:

- Accession codes, unique identifiers, or web links for publicly available datasets
- A description of any restrictions on data availability
- For clinical datasets or third party data, please ensure that the statement adheres to our policy

ChIP-Seq and RNA-Seq data are aligned to mm10 mouse genome, and are deposited at GEO with accession numbers GSE255466, GSE255562, GSE255563 and GSE25587.

## Human research participants

Policy information about studies involving human research participants and Sex and Gender in Research.

| Reporting on sex and gender | N/A |
|---|---|
| Population characteristics | N/A |
| Recruitment | N/A |
| Ethics oversight | N/A |

Note that full information on the approval of the study protocol must also be provided in the manuscript.

# Field-specific reporting

Please select the one below that is the best fit for your research. If you are not sure, read the appropriate sections before making your selection.

☒ Life sciences          ☐ Behavioural & social sciences          ☐ Ecological, evolutionary & environmental sciences

For a reference copy of the document with all sections, see nature.com/documents/nr-reporting-summary-flat.pdf

# Life sciences study design

All studies must disclose on these points even when the disclosure is negative.

| Sample size | No statistical methods were used to predetermine the number of animals and cells. Suitable sample sizes were estimated based on previous published reports (Donato et al., Nature, 2013, Dehorter et al., Science, 2015, Mauger et al., Neuron, 2016, Xiao et al., Nature Communications, 2018, Hörnberg et al., Nature, 2020). |
|---|---|
| Data exclusions | Mice were excluded for CNO experiments when the expression of hM4Di virus was not sufficiently spread or if the mice had to be euthanized prematurely due to severe seizures. For patch clamp experiments, data was excluded if the recorded cells displayed >20% change in the series resistance during recordings. |
| Replication | All animal experiments were performed with animals from several litters. Each cohort showed similar phenotypes.<br>For immunostaining, western blot and FISH experiments, a minimum of 3 animals per genotype was used. For cell culture experiments, a minimum of 3 biological replicates were used and no attempts of experiments were excluded. N numbers are provided for each experiment in |

| | the figure legends. |
|---|---|
| Randomization | Animals were randomly assigned to treatment groups at the time of viral injections. Cultures were also randomly assigned at the time of experiments. |
| Blinding | Experimenter was blinded to genotype for all experiments, and most analysis. For electrophysiological experiments, the experimenter was blinded for genotype. For EEG recordings, blinding for analysis was limited as the analysis was selective to identify the epileptiform brain activity. However, assessment for seizure detection and analysis were done by different experimenters, thereby enabling some blinding to the genotype. |

# Reporting for specific materials, systems and methods

We require information from authors about some types of materials, experimental systems and methods used in many studies. Here, indicate whether each material, system or method listed is relevant to your study. If you are not sure if a list item applies to your research, read the appropriate section before selecting a response.

## Materials & experimental systems

| n/a | Involved in the study |
|---|---|
| ☐ | ☒ Antibodies |
| ☐ | ☒ Eukaryotic cell lines |
| ☒ | ☐ Palaeontology and archaeology |
| ☐ | ☒ Animals and other organisms |
| ☒ | ☐ Clinical data |
| ☒ | ☐ Dual use research of concern |

## Methods

| n/a | Involved in the study |
|---|---|
| ☐ | ☒ ChIP-seq |
| ☒ | ☐ Flow cytometry |
| ☒ | ☐ MRI-based neuroimaging |

# Antibodies

| Antibodies used | Primary antibodies:<br>rabbit Smad1, Cell Signaling, 6944, lot #6, 1to100 for ChIP and 1to1000 for WB<br>rabbit Smad5, Cell Signaling, 12534, lot #2, 1to100 for ChIP and 1to1000 for WB<br>rabbit-anti-phospho-SMAD1/5/9, Cell Signaling, 13820, lot #3, 1to800 for ICC and 1to1000 for WB<br>rabbit-anti-H3K27ac, Abcam 4729, lot #GR3231988-1, 1to1000<br>mouse-anti-BMPR2, BD Pharmingen, 612292, lot #7131991, 1to1000<br>rabbit-anti-Calnexin, stressGen, SPA-865, lot #11041924, 1to2000<br>rabbit-anti-HA, Cell Signaling, 3724, lot #10, 1to1000<br>rat-anti-GAPDH, Biolegend, 607902, lot #B259205, 1to10000<br>mouse-anti-MAP2, Synaptic systems, 188011, lot #1-17, clone 198A5, 1to1000<br>mouse-anti-CamKII alpha, ThermoFisher, MA1-048, lot #TH269517, 1to800<br>mouse-anti-GAD67, Millipore, MAB5406, lot #53872310, clone #1G10.2, 1to500<br>rabbit-anti NeuN, Abcam ab177482, lot#1001571-1, 1to500<br>goat anti-Parvalbumin antibody, Swant, PVG214, RRID: AB_10000345, 1to5000<br>biotinylated WFA, Vector laboratories, B-1355-2, lot #SLBQ2585V, 1to500<br>Monoclonal mouse-anti-Synaptotagmin 2, Zebrafish International Resource Center, ZNP-1, RRID: AB_10013783, 1to1000<br>rabbit-anti-vGlut1 polyclonal purified antibody, Synaptic Systems, 135303, lot #4-90, 1to5000<br>mouse anti-GFP, Santa Cruz, sc-9996, 1to1000<br><br>Secondary antibodies<br>goat anti-rat-HRP, Jackson ImmunoResearch, 112-035-143, lot #152008, 1to10000<br>goat-anti-mouse-HRP, Jackson ImmunoResearch, 115-035-149, lot #120343, 1to10000<br>goat anti-rabbit-HRP, Jackson ImmunoResearch, 111-035-003, 1to10000<br>Cy3-conjugated donkey anti-mouse, Jackson ImmunoResearch, 715-165-151, lot #164090, 1to500<br>Cy3-conjugated donkey anti-rabbit, Jackson ImmunoResearch, 711-165-152, lot #160467, 1to500<br>Cy5-conjugated donkey anti goat, Jackson ImmunoResearch, 705-175-147, lot #818817, 1to500<br>Alexa405 goat anti-rabbit (Thermo Scientific #A-31556), 1to500<br>Alexa 488 conjugated donkey anti rabbit, ThermoFisher, R37118, 1to1000<br>Alexa 647 conjugated donkey anti mouse, Jackson ImmunoResearch, 715-605-151, lot #140647, 1to1000<br>Cy2-conjugated Streptavidin, Jackson ImmunoResearch, 016-220-084, 1to1000<br>Alexa 647 conjugated steptavidin, Thermo Fisher, A31556, lot # 1010119, 1to1000<br>Cy5-conjugated donkey-anti-rabbit, Jackson ImmunoResearch, 711-175-152, lot #161788, 1to500<br>Cy5 donkey anti-mouse, Jackson #715-175-511, Lot #165683, 1to500<br>Cy3-conjugated donkey-anti-guineapig, Jackson ImmunoResearch, 706-165-148, lot #154462, 1to500 |
|---|---|
| Validation | Smad1/Smad5: WB: validated using various cell cell lines. ChIP: Chromatin immunoprecipitations were performed with cross-linked chromatin from MCF7 cells treated with Human BMP2 for one hour and either Smad1 Rabbit mAb or Normal Rabbit IgG (cell.signal.com)<br>H3K27ac: ChIP: Chromatin immunoprecipitations were performed with cross-linked chromatin from HeLa cells (abcam.com)<br>pSmad1/5/9: IHC: validated from HT-1080 cells, serum-starved (overnight; left) or serum-starved and treated with Human BMP2 #4697 (50 ng/ml, 30 min; right)<br>GAPDH: validated by western blotting using extracts from  K562, PC3, HeLa, Molt-4, NTRA-2, NIH3T3, and UMR-106 cells |

(biolegend.com)
Calnexin: validated by western blotting using extracts from MWM, Vero, 3T3, PC-12 and HeLa cells (enzolifesciences.com)
BMPR2: validated by western blotting using pulmonary artery smooth muscle cell extract (labome.com)
HA: validated by western blotting of extracts from HeLa cells, untransfected or transfected with either HA-FoxO4 or HA-Akt3 (cellsignal.com)
MAP2: validated for immunocytochemistry from primary rat hippocampal neurons (sysy.com)
anti CamKII alpha: validated for immunocytochemistry from primary mouse cortical cultures (thermofisher.com)
anti-GAD67: validated for immunohistochemistry from mouse and rat tissue (merckmillipore.com)
anti-NeuN: validated for immunohistochemistry from mouse and human tissue (abcam.com)
anti-Parvalbumin antibody: Validated for immunohistochemistry from parvalbumin knock-out mouse cortex and hippocampus tissues (swant.com)
anti-GFP antibody: validated by western blotting using extracts from COS cells, untransfected or transfected with GFP (scbt.com)
biotinylated WFA: validated for immunohistochemistry from octadon degu Alzheimer model (vectorlabs.com)
Monoclonal mouse-anti-Synaptotagmin 2: validated for immunohistochemistry in brain tissues (zfin.org)
rabbit-antivGlut1 polyclonal purified antibody: validated for immunohistochemistry by staining of hippocampus sections from mouse (sysy.com)

## Eukaryotic cell lines

Policy information about cell lines and Sex and Gender in Research

| | |
|---|---|
| Cell line source(s) | Primary cortical cells were prepared from the 16.5 days old embryos of RjOrl:SWISS mice (Janvier) mice or P0 pups of C57BL/6j mice. AAVpro HEK293T cell line was obtained from TakaraBio (#632273) and used at passage numbers from 8-20. |
| Authentication | Cell lines evaluated by observing their morphology and growth. |
| Mycoplasma contamination | Regular inspection of cell health and survival didn't show any indication for mycoplasma contamination. |
| Commonly misidentified lines (See ICLAC register) | no commonly misidentified lines were used in this study. |

## Animals and other research organisms

Policy information about studies involving animals; ARRIVE guidelines recommended for reporting animal research, and Sex and Gender in Research

| | |
|---|---|
| Laboratory animals | Mice:<br>C57BL/6j, both males and females, P0 to adult (8-16 weeks)<br>RjOrl:SWISS mice (Janvier),male, female, E16.5 |
| Wild animals | the study did not involve wild animals |
| Reporting on sex | Findings were obtained from both males and females. Sex was assigned by trained caretakers before toe marking for genotyping or at the weaning age. For in vitro experiments cells were isolated from embryos or P0 pups, sex information was not collected. For other experiments, matching number of animals for males and females were used. |
| Field-collected samples | the study did not involve field-collected samples |
| Ethics oversight | Basel Cantonal Veterinary Office Committees for Animal Experimentation |

Note that full information on the approval of the study protocol must also be provided in the manuscript.

## ChIP-seq

### Data deposition

☒ Confirm that both raw and final processed data have been deposited in a public database such as GEO.

☒ Confirm that you have deposited or provided access to graph files (e.g. BED files) for the called peaks.

| | |
|---|---|
| Data access links<br>*May remain private before publication.* | https://www.ncbi.nlm.nih.gov/geo/query/acc.cgi?acc=GSE255466<br>https://www.ncbi.nlm.nih.gov/geo/query/acc.cgi?acc=GSE255587 |
| Files in database submission | smadsample1_R1.fastq.gz<br>smadsample1_R2.fastq.gz<br>smadsample2_R1.fastq.gz<br>smadsample2_R2.fastq.gz<br>smadsample3_R1.fastq.gz<br>smadsample3_R2.fastq.gz<br>smadsample4_R1.fastq.gz<br>smadsample4_R2.fastq.gz<br>input_cneg1_R1.fastq.gz<br>input_cneg1_R2.fastq.gz |

```
input_cneg2_R1.fastq.gz
input_cneg2_R2.fastq.gz
input_cneg3_R1.fastq.gz
input_cneg3_R2.fastq.gz
input_cneg4_R1.fastq.gz
smad_vs_input_rep1_peaks.narrowPeak
smad_vs_input_rep2_peaks.narrowPeak
smad_vs_input_rep3_peaks.narrowPeak
smad_vs_input_rep4_peaks.narrowPeak
Bmp2_Input_rep1_S1_R1_001_MM_1.fastq.gz
Bmp2_Smad1_5_rep1_S2_R1_001_MM_1.fastq.gz
Bmp2_H3K27ac_rep1_S2_R1_001_MM_1.fastq.gz
NT_Input_rep1_S1_R1_001_MM_1.fastq.gz
NT_Smad1_5_rep1_S2_R1_001_MM_1.fastq.gz
NT_H3K27ac_rep1_S2_R1_001_MM_1.fastq.gz
Bmp2_Input_rep2_S1_R1_001_MM_1.fastq.gz
Bmp2_Smad1_5_rep2_S2_R1_001_MM_1.fastq.gz
Bmp2_H3K27ac_rep2_S2_R1_001_MM_1.fastq.gz
NT_Input_rep2_S1_R1_001_MM_1.fastq.gz
NT_Smad1_5_rep2_S2_R1_001_MM_1.fastq.gz
NT_H3K27ac_rep2_S2_R1_001_MM_1.fastq.gz
Bmp2_Input_rep3_S1_R1_001_MM_1.fastq.gz
Bmp2_Smad1_5_rep3_S2_R1_001_MM_1.fastq.gz
Bmp2_H3K27ac_rep3_S2_R1_001_MM_1.fastq.gz
NT_Input_rep3_S1_R1_001_MM_1.fastq.gz
NT_Smad1_5_rep3_S2_R1_001_MM_1.fastq.gz
NT_H3K27ac_rep3_S2_R1_001_MM_1.fastq.gz
Bmp2_Input_rep1_S1_R2_001_MM_1.fastq.gz
Bmp2_Smad1_5_rep1_S2_R2_001_MM_1.fastq.gz
Bmp2_H3K27ac_rep1_S2_R2_001_MM_1.fastq.gz
NT_Input_rep1_S1_R2_001_MM_1.fastq.gz
NT_Smad1_5_rep1_S2_R2_001_MM_1.fastq.gz
NT_H3K27ac_rep1_S2_R2_001_MM_1.fastq.gz
Bmp2_Input_rep2_S1_R2_001_MM_1.fastq.gz
Bmp2_Smad1_5_rep2_S2_R2_001_MM_1.fastq.gz
Bmp2_H3K27ac_rep2_S2_R2_001_MM_1.fastq.gz
NT_Input_rep2_S1_R2_001_MM_1.fastq.gz
NT_Smad1_5_rep2_S2_R2_001_MM_1.fastq.gz
NT_H3K27ac_rep2_S2_R2_001_MM_1.fastq.gz
Bmp2_Input_rep3_S1_R2_001_MM_1.fastq.gz
Bmp2_Smad1_5_rep3_S2_R2_001_MM_1.fastq.gz
Bmp2_H3K27ac_rep3_S2_R2_001_MM_1.fastq.gz
NT_Input_rep3_S1_R2_001_MM_1.fastq.gz
NT_Smad1_5_rep3_S2_R2_001_MM_1.fastq.gz
NT_H3K27ac_rep3_S2_R2_001_MM_1.fastq.gz
Bmp2_Input_1.bw
Bmp2_Smad1_5_1.bw
Bmp2_H3K27ac_1.bw
NT_Input_1.bw
NT_Smad1_5_1.bw
NT_H3K27ac_1.bw
Bmp2_Input_2.bw
Bmp2_Smad1_5_2.bw
Bmp2_H3K27ac_2.bw
NT_Input_2.bw
NT_Smad1_5_2.bw
NT_H3K27ac_2.bw
Bmp2_Input_3.bw
Bmp2_Smad1_5_3.bw
Bmp2_H3K27ac_3.bw
NT_Input_3.bw
NT_Smad1_5_3.bw
NT_H3K27ac_3.bw
```

Genome browser session
(e.g. UCSC)

N/A

## Methodology

Replicates

3 biological replicates that were generated from independent litters were used for in vitro ChIP-seq experiments. 4 biological replicates (2 males and 2 females) were used for in vivo ChiP-seq experiments.

Sequencing depth

The sequencing was performed paired end 41 bases yielded around 50±2 millions pass filter reads for in vitro and 19-32 millions pass filter reads for in vivo samples..

| | |
|---|---|
| Antibodies | rabbit anti-Smad1/Smad5 (Cell Signaling #6944 and #12534) and rabbit anti-H3K27ac (Abcam 4729) antibodies were used. |
| Peak calling parameters | Peaks were called for each ChIP replicate against a matched input using the MACS2 callpeak function with the default options. |
| Data quality | Peaks existed in at least 2 out of 3 replicates for each group were considered which yielded in total 896 peaks. For comparisons between control and Bmp2 treated conditions, peaks that had at least 1.4 fold change and less then 0.05 adjusted p-value were considered as significant. For in vivo ChiP-seq, peaks existed in at least 3 out of 4 replicates were considered which yielded 239 peaks. |
| Software | MACS2 and R softwares were used by using default parameters for analysis. |

