## [Peer Review File · Nature]

Manuscript Title: Control of neuronal excitation - inhibition balance by BMP-SMAD1 signaling

Reviewer Comments & Author Rebuttals

Reviewer Reports on the Initial Version:

Referees' comments:

Referee #1 (Remarks to the Author):

In this study, Okur and colleagues investigate the role of BMP-SMAD1 signaling in the regulation of excitation-inhibition balance in the brain. BMPs are classical morphogens that are essential during early mammalian development and surprisingly, the authors find that this signaling system is co-opted in adulthood to regulate the activity and connectivity of PV interneurons in maintaining network stability. BMP2 is upregulated and secreted by neuronal cultures upon increased neuronal activity. Chemogenetic inhibition of PV interneurons to increase network activity upregulates BMP signaling in the form of increased target gene expression and the activation of a BMP-dependent synthetic reporter in PV interneurons. CHIP-seq and RNA-seq of BMP2-stimulated neuronal cultures show that SMAD1/5 binds target gene promoters leading to upregulation of genes involved in glutamatergic transmission and extracellular matrix in a SMAD1-dependent manner. Smad1 conditional KO (cKO) in PV interneurons results in decreased glutamatergic innervation of PV interneurons, decreased firing rates, decreased Parvalbumin and PNN staining intensity, and ultimately an imbalance of excitatory/inhibitory balance that can result in spontaneous seizures.

This is an impressive manuscript that identifies a new pathway for the maintenance of cortical network homeostasis, reveals the repurposing of developmental morphogens for the control of excitation/inhibition balance in circuits in adulthood, and has potential implications for neurodevelopmental disorders. Experiments are well designed and multiple lines of evidence are used to corroborate the paper's findings. The paper is well written and the quality of the figures is high. Some clarifications are needed, as outlined below, but these comments do not diminish my overall enthusiasm for this study.

Major Points:

1. Cell type specificity of BMP2-SMAD1 signaling: Ext. Data Fig.1c shows upregulation of Bmp2 expression in glutamatergic neurons upon increased activity. Is Bmp2 expression also upregulated in inhibitory neurons in the cortical culture upon bicuculine treatment? Fig.2 uses a mix of excitatory and inhibitory neurons in culture suggesting SMAD signaling can be activated in all cell types, as also shown in Ext. Fig.4a-c. What would be the role of BMP-SMAD1 signaling in excitatory neurons? The transcellular communication pathway of glutamatergic neuron-derived BMP2 to PV interneuron SMAD signaling is a major claim the authors make in the manuscript; it is therefore important to show that BMP2 from glutamatergic neurons and not interneurons induces SMAD signaling in PV interneurons, or adjust this statement.
2. The 4xBMP-response element sequence from the Id1 promoter was used before (Lewis and Prywes, BBA 2013) and is elegantly combined here with the small molecule-gated miniXon. The nuclear reporter is dependent on the presence of the small molecule (Ext. Data Fig.3), but it is not shown that the reporter signal is BMP-dependent. This should be tested, using the Noggin inhibitor in an in vitro assay.
3. The authors conclude that upon increased neuronal activity, BMP2-SMAD1 signaling provides a trans-neuronal signal that adjusts functional PV interneuron recruitment to maintain network excitation-inhibition balance. The Smad1 loss of function experiments are performed using Pv-Cre mice, and developmental defects from postnatal Smad1 deletion can therefore not be excluded. A

more convincing demonstration of the model the authors propose would be to experimentally elevate activity in adult cortex (for instance using the chemogenetic approach of Fig.1b) and combine this with acute loss of Smad1 function in PV interneurons. This would then be expected to prevent an activity-induced increase in excitatory drive of PV interneurons and provide proof that BMP2-SMAD1 signaling regulates network homeostasis in adult neural circuits. An alternative would be to remove Smad1 in adult PV interneurons and assess the same parameters on innervation and function of PV cells (Fig.3) after an equivalent amount of time. Another alternative approach would be to demonstrate absence of innervation defects in young (P26-30) Pv-Cre:Smad1 cKO animals using the FingR approach and mE/IPSC recordings.

Minor Points:

1. Pg. 2 "...the canonical BMP-target genes Id1 and Smad6 were significantly upregulated in stimulated neocortical cultures." Ext. Data Fig.1i shows that Id1 and Id3 are significantly upregulated but not Smad6. Please correct.
2. Ext. Data Fig.2b: The color coding of the legends does not match the box colors in the plot, which makes it difficult to interpret the data.
3. Fig.1c: Only Id1 and Smad6 are significantly upregulated compared to saline control, please correct statement in the text (pg. 3).
4. Bcan does not show a significant increase after BMP2 treatment using qPCR analysis (Fig.2f). This is unexpected for a BMP-SMAD1 transcriptional target and suggests the link between BMP-SMAD1 signaling and PNNs is less direct than the authors claim. Please discuss.
5. The authors mention "Differential gene expression analysis identified 30 and 147 up-regulated transcripts 1 and 6 hours after BMP2-stimulation, respectively (Extended Data Fig. 4c, Supplementary Table 2)." I think the authors mean to refer to Ext. Fig. 4d instead of 4c in this context.
6. Ext. Fig. 4f: I found this graph difficult to understand, please explain the experiment in more detail. How do the authors explain the significant increase in constitutive SMAD1 regulated genes?
7. Fig.2b-d: The authors mention that "constitutively bound regions exhibited only low H3K27ac signal (Fig. 2b, c) suggesting that they are transcriptionally silent." It is unclear what the authors mean here. Constitutive sites appear to still contain H3K27ac as shown in Fig.2b,c. Please clarify. The authors do not discuss the motif of the constitutive binding site (Fig.2d). As SMAD1/5 is constitutively bound to these sites, is this also a known motif for SMAD1/5 binding?
8. The authors conclude that "SMAD1 is the key downstream mediator of BMP signaling in mature neurons and its neuronal loss of function results in a severe imbalance of neuronal network activity in vitro." (pg. 4). This claim is overstated here and should be adjusted. The authors show that SMAD1 neuronal loss of function results in gene deregulation including alteration of activity-regulated rPRGs and SRGs. However, a change in rPRGs and SRG levels does not necessarily result in an imbalance in neuronal network activity, since these gene expression alterations take place during various neuronal activity patterns that may underlie normal neuronal network activity (Tyssowski et al., Neuron 2018).
9. Fig 3 and Ext. Fig.6: why did the authors end up choosing PSD.FingR over Xph20? Different probes are shown in Ext. Fig.6a but the choice was not motivated.
10. The authors do not discuss the observed electrophysiological differences between young (P26-30) and adult (P56-72) mice upon SMAD1 deletion. Could the authors provide an explanation for this observed difference?

Referee #3 (Remarks to the Author):

This is a well written manuscript that investigates the role of BMP2-SMAD1 signaling towards synaptic connectivity into PV cells using slice e-phys. The study further shows that conditional loss of Smad1 in PV cells leads to seizures, but is not clear if seizure activity results from reduced inhibition from cortical PV cells, or if changes in cortical PV cells are the result of seizure activity that can be initiated in any brain region. The study is of moderate interest and perhaps more

suitable for a specialized journal.

Major points:

1. Histology is not a quantitative way to quantify protein expression. The increase in BMP2 using histology (Fig. Fig 1F-G) should be addressed using a quantitative method such as western blots. Same goes for other similar quantifications such as PV quantification in Fig 4. Analyzing the integrity of WFAs would be more informative than WFA intensity.

2. Figure 2 is entirely in cell culture, a system very different from the intact brain. It is unclear of similar results would be observed in the intact brain, a critical point for this study.

3. Is adult BMP2-SMAD1 signaling specific for PV cells or is it a general phenomena? The authors show loss of Smad1 in PV cells leads to reduced mEPSC frequency into those cells, but the study was restricted to PV-Cre mice, and there is therefore no reason to think that the phenotypes observed in PV cells is not generalizable to other cells types had them be analyzed.

4. Changes in PV cells activity might be due the result of seizures rather than the other way around making the conclusions that we can drive from Figure 4 ambiguous.

Author Rebuttals to Initial Comments: Response to Reviewers

We thank the reviewers for their thoughtful, detailed, and constructive comments on the manuscript. This feedback encouraged us to further advance our genetic marking and loss-of-function tools and to carry out several new experiments. We feel that with these revisions, the manuscript has been substantially improved and we hope that the reviewers would consider the work now suitable for publication in *Nature*.

Below, we provide a point-by-point response to all comments made by the reviewers (comments in black, answers in blue).

Referee #1

Major Points:

1. Cell type specificity of BMP2-SMAD1 signaling: Ext. Data Fig.1c shows upregulation of *Bmp2* expression in glutamatergic neurons upon increased activity. Is *Bmp2* expression also upregulated in inhibitory neurons in the cortical culture upon bicuculine treatment? Fig.2 uses a mix of excitatory and inhibitory neurons in culture suggesting SMAD signaling can be activated in all cell types, as also shown in Ext. Fig.4a-c. What would be the role of BMP-SMAD1 signaling in excitatory neurons? The transcellular communication pathway of glutamatergic neuron-derived BMP2 to PV interneuron SMAD signaling is a major claim the authors make in the manuscript; it is therefore important to show that BMP2 from glutamatergic neurons and not interneurons induces SMAD signaling in PV interneurons, or adjust this statement.

The reviewer lists multiple questions that all concern the origin of the BMP2 ligand and the identity of the recipient cells for the BMP signals. We performed two new genetic experiments to address this:

- a) To directly test whether BMP2 derived from glutamatergic neurons in vivo is important, we generated an adult conditional knock-out of BMP2 from upper layer glutamatergic cells using the *Cux2*CreERT2 driver *Bmp2*^{Δ*Cux2*}. We then developed a new cre-independent version of the FingR-PSD95 probe (using the S5E2 enhancer) to be able to visualize glutamatergic synapses in parvalbumin interneurons of *Bmp2*^{Δ*Cux2*} mice. We find that selective loss of BMP2 ligand from glutamatergic neurons indeed results in a reduction in glutamatergic synapse density on PV interneurons. These results are now included in the **new Figure 3** of the revised manuscript.
- b) As pointed out by the reviewer, we find that in vitro, glutamatergic and GABAergic neurons are competent of BMP-SMAD1 signaling. To test which cell populations besides PV interneurons activate BMP-SMAD1 signaling in response to elevated network activity in vivo, we generated a pan-neuronal form of the BRX-reporter and tested it in the chemogenetic stimulation experiments. We find that PV interneurons are the population with the highest reporter activity and the only population with significant elevation of reporter activity in response to stimulation. This does not exclude functions for BMP-SMAD1 signaling in other neuronal cells but highlights that its most significant activation in response to neuronal activity occurs in PV interneurons. These **new results** are included in **Fig.1d-g**.

2. The 4xBMP-response element sequence from the *Id1* promoter was used before (Lewis and Prywes, BBA 2013) and is elegantly combined here with the small molecule-gated miniXon. The nuclear reporter is dependent on the presence of the small molecule (Ext. Data Fig.3), but it is not shown that the reporter signal is BMP-dependent. This should be tested, using the Noggin inhibitor in an in vitro assay.

Considering that reviewer 2 strongly recommended to limit in vitro experimentation, we further probed BMP-dependence of the reporter in vivo. We developed a doxycycline-inducible, constitutively active BMP-receptor expression vector and activated the BMP-signaling pathway selectively in PV interneurons in the barrel cortex. We observed that this resulted in strong activation of the BRX-reporter

in PV interneurons, thus, providing evidence for the BMP-dependency of the reporter *in vivo*. This new data is included in the **new Extended Data Figure 4**.

3. The authors conclude that upon increased neuronal activity, BMP2-SMAD1 signaling provides a trans-neuronal signal that adjusts functional PV interneuron recruitment to maintain network excitation-inhibition balance. The Smad1 loss of function experiments are performed using Pv-Cre mice, and developmental defects from postnatal Smad1 deletion can therefore not be excluded. A more convincing demonstration of the model the authors propose would be to experimentally elevate activity in adult cortex (for instance using the chemogenetic approach of Fig.1b) and combine this with acute loss of Smad1 function in PV interneurons. This would then be expected to prevent an activity-induced increase in excitatory drive of PV interneurons and provide proof that BMP2-SMAD1 signaling regulates network homeostasis in adult neural circuits. An alternative would be to remove Smad1 in adult PV interneurons and assess the same parameters on innervation and function of PV cells (Fig.3) after an equivalent amount of time. Another alternative approach would be to demonstrate absence of innervation defects in young (P26-30) Pv-Cre:Smad1 cKO animals using the FingR approach and mE/IPSC recordings.

We currently do not have the genetic tools in the laboratory to ablate SMAD1 selectively in PV cells in adult mice. We attempted to generate a novel viral tool by combining PV cell-specific enhancer elements and a GABA neuron-specific splicing-dependent frameshift design (building on Ling et al., *Nature Communications* 2022). However, despite the combination of two genetic control elements that should be interneuron specific, the resulting recombination observed with this tool was not PV interneuron-specific.

To overcome this problem, we used adult ablation of BMP2 in glutamatergic cells (with the Cux2CreERT2 driver which we did have available in the lab) as an alternative strategy for adult loss-of-function. As described above in the answer to point 1 we observed that adult ablation of BMP2 resulted in reduced glutamatergic synapse density onto PV interneurons (**new Figure 3**), thus, supporting an adult function, rather than developmental role for the BMP-SMAD1 signaling pathway.

We further used temporally-regulated expression of constitutively active BMPRIIA receptor in adult PV interneurons to further interrogate the outcome of BMP-SMAD1 signaling in the adult barrel cortex. We find that activation of the pathway in the adult is sufficient to increase parvalbumin and WFA staining intensity. This provides additional evidence supporting an adult function (rather than developmental defects) resulting from the manipulation of the pathway. These new results are included in **new Extended Data Figure 8**.

As a complementary effort (and as suggested by the reviewer), we examined potential developmental effects of *Smad1* ablation in younger mice. We performed mEPSC recordings from *Smad1^{ΔPV}* mice at postnatal day 26-30 and found no change in mEPSC frequency and amplitude, suggesting that the glutamatergic innervation of PV cells during development proceeds normally in the absence of BMP-SMAD1 signaling. This new data is included in the **new Extended Data Figure 9c-e**.

Minor Points:

1. Pg. 2 ‘...the canonical BMP-target genes Id1 and Smad6 were significantly upregulated in stimulated neocortical cultures.’: Ext. Data Fig.1i shows that Id1 and Id3 are significantly upregulated but not Smad6. Please correct.

Thank you - done

2. Ext. Data Fig.2b: The color coding of the legends does not match the box colors in the plot, which makes it difficult to interpret the data.

We have now clarified the labeling.

3. Fig.1c: Only Id1 and Smad6 are significantly upregulated compared to saline control, please correct statement in the text (pg. 3).

Thank you – done

4. *Bcan* does not show a significant increase after BMP2 treatment using qPCR analysis (Fig.2f). This is unexpected for a BMP-SMAD1 transcriptional target and suggests the link between BMP-SMAD1 signaling and PNNs is less direct than the authors claim. Please discuss.

Bcan was identified as significantly up-regulated by RNA-Seq (data in Extended Data Figure 5d) and was identified as a directly bound gene in the SMAD1 ChIP experiments (Fig.2e). *Bcan* shows a highly significant reduction in basal expression in *Smad1* conditional knock-out neurons as assessed by qPCR (Fig.2f). However, as remarked by the reviewer, the figure with the qPCR-based assays reported only a trend for *Bcan* upregulation in response to BMP2 stimulation ($p=0.107$). Given that three out of 5 samples, showed a strong upregulation of *Bcan* mRNA 6 hrs after stimulation and two additional samples exhibited a slight increase we re-checked the calculation of the statistical significance. It turns out that the appropriate p-value was 0.0031. We apologize for this error and thank the reviewer for pointing this out!

5. The authors mention “Differential gene expression analysis identified 30 and 147 up-regulated transcripts 1 and 6 hours after BMP2-stimulation, respectively (Extended Data Fig. 4c, Supplementary Table 2).” I think the authors mean to refer to Ext. Fig. 4d instead of 4c in this context.

Thank you – we corrected this

6. Ext. Fig. 4f: I found this graph difficult to understand, please explain the experiment in more detail. How do the authors explain the significant increase in constitutive SMAD1 regulated genes?

We now expanded the explanation in the figure legend. The simplest explanation for the increase in gene expression from constitutively-bound genes is that SMAD1 (in combination with interaction partners) acts as a transcriptional repressor on these genes, a function that has also been observed in developmental studies on SMAD1.

7. Fig.2b-d: The authors mention that “constitutively bound regions exhibited only low H3K27ac signal (Fig. 2b, c) suggesting that they are transcriptionally silent.” It is unclear what the authors mean here. Constitutive sites appear to still contain H3K27ac as shown in Fig.2b,c. Please clarify. The authors do not discuss the motif of the constitutive binding site (Fig.2d). As SMAD1/5 is constitutively bound to these sites, is this also a known motif for SMAD1/5 binding?

Overall, the H3K27ac signal was lower on the constitutively-bound sites (see Fig.2c). However, we agree with the reviewer that the statement that these sites are possibly “silent” is mis-leading. We now corrected this and also added information on the motif of the constitutive binding sites.

8. The authors conclude that “SMAD1 is the key downstream mediator of BMP signaling in mature neurons and its neuronal loss of function results in a severe imbalance of neuronal network activity in vitro.” (pg. 4). This claim is overstated here and should be adjusted. The authors show that SMAD1 neuronal loss of function results in gene deregulation including alteration of activity-regulated rPRGs and SRGs. However, a change in rPRGs and SRG levels does not necessarily result in an imbalance in neuronal network activity, since these gene expression alterations take place during various neuronal activity patterns that may underlie normal neuronal network activity (Tyssowski et al., Neuron 2018).

We have adjusted the statement in the text.

9. Fig 3 and Ext. Fig.6: why did the authors end up choosing PSD.FingR over Xph20? Different probes are shown in Ext. Fig.6a but the choice was not motivated.

In PV interneurons of adult mice, the FingR-PSD-95 probe produced labeling with superior synaptic over non-synaptic labeling as compared to Xph20. We have now explicitly stated this in the legend (now Extended Data Fig. 6).

10. The authors do not discuss the observed electrophysiological differences between young (P26-30) and adult (P56-72) mice upon SMAD1 deletion. Could the authors provide an explanation for this

observed difference?

We have now commented on this in the discussion (page 6).

Referee #3

Major points:

1. Histology is not a quantitative way to quantify protein expression. The increase in BMP2 using histology (Fig. Fig 1F-G) should be addressed using a quantitative method such as western blots. Same goes for other similar quantifications such as PV quantification in Fig 4. Analyzing the integrity of WFAs would be more informative than WFA intensity.

We agree with the reviewer that quantitative histology is challenging and dependent on high quality immunological reagents. The experiments in Fig. 1F-G (now new Fig. 1h and i) do not rely on antibodies but on the fluorescence of a GFP reporter that enables quantification of BMP-signaling in situ (similar to widely used and accepted tools to read-out cFos induction (Sorensen *et al*, 2016; Yap *et al*, 2021) or calcium transients (Zhang *et al*, 2023). That means it is not confounded with the problems of antibody penetration and specificity that can plague histology approaches. The increase in BMP2 expression was shown in Expanded Data Figure 1c and was assessed by single molecule fluorescence in situ hybridization, a method that is widely accepted to yield highly quantitative measures of mRNA abundance in situ.

The quantification of parvalbumin protein intensity shown in Figure 5b and Extended Data Figure 7 are indeed antibody-based. We note that this is a widely used method for quantification [e.g. see reference (Donato *et al*, 2013) or (Ramsaran *et al*, 2023)]. Regarding the use of Western blots for quantification, we note that our histology data indicates that only PV interneurons in layer 2/3 exhibit a significant reduction in parvalbumin immune-reactivity. However, the majority of parvalbumin interneurons are found in deeper layers. Thus, a method for local measurement of PV intensity is necessary and, unfortunately, this cannot be replaced by using western blots or conventional proteomic methods. As for the analysis of the integrity of WFA-labeled PNNs we performed the suggested analysis. We find that PNN integrity is indeed compromised in *Smad1^{ΔPV}* mice. The new data is provided in **Extended Data Figure 7**.

2. Figure 2 is entirely in cell culture, a system very different from the intact brain. It is unclear of similar results would be observed in the intact brain, a critical point for this study.

There are two reasons for conducting these experiments in a primary culture preparation: a) antibody tools for ChIP or cut & run analyses are often rate-limiting and only a small fraction of these reagents have sufficiently high binding affinity to produce high quality data when directly applied on brain samples. b) Transcriptional responses to BMP signaling occur rapidly and activate repressors that terminate signaling. This inhibitory feedback loop complicates identifying transcriptional targets in systems where signaling is not initiated acutely and simultaneously in all cells. This is why genome-wide mapping of transcriptional signaling targets is typically conducted in culture where signaling can be temporally controlled.

Regardless, we now carried out ChIP-Seq experiments with adult somatosensory cortex tissue. We find that 42.5% of the constitutive binding events mapped in vitro are also mapped as significant binding events in the intact brain preparation and conversely 77% of binding events detected in vivo are identified as significant in vitro. This is consistent with the (expected) higher sensitivity of the in vitro experiment and confirms that in vivo binding events are replicated under the in vitro conditions. This **new data** supports the validity of our model system and is now included in **Figure 2**.

Given that Figure 2 uses recombinant BMP2 for stimulation, we also sought to test the role of endogenous BMP2 ligands in controlling PV interneuron innervation. We generated conditional knock-out mice where BMP2 is selectively ablated in adult upper layer principal neurons (*Cux2^{CreERT2}::BMP2^{fl/fl}*) with tamoxifen-induced recombination induced in 5-6 week old mice. We find that BMP2 ablation in glutamatergic cells results in a reduction in glutamatergic synapses formed onto

PV interneurons (**new Figure 3**). This provides genetic in vivo evidence for a BMP-ligand-dependent regulation of PV cell function.

3. Is adult BMP2-SMAD1 signaling specific for PV cells or is it a general phenomenon? The authors show loss of Smad1 in PV cells leads to reduced mEPSC frequency into those cells, but the study was restricted to PV-Cre mice, and there is therefore no reason to think that the phenotypes observed in PV cells is not generalizable to other cell types had them be analyzed.

To address this point, we generated a “global” (not cell type-specific) BMP signaling reporter, introduced it into the adult barrel cortex, and then chemogenetically elevated neuronal network activity. By comparing the activation of the BMP reporter across cell types, we observed that the BMP-signaling pathway is most strongly activated in parvalbumin interneurons (**new Figure 1d-g**). This provides strong rationale for examining the consequences of *Smad1* ablation specifically in these cells.

The purpose of analyzing PV cell-specific conditional knock-outs was to specifically test the hypothesis that the excitation – inhibition balance phenotype results from activation of SMAD1 in PV cells. SMAD1 is expressed in other neuron populations. We hope that the reviewer would agree that a broader loss-of-function model would be more difficult to interpret with respect to the mechanism.

It is conceivable that the function of BMP2-SMAD1 signaling discovered in this study might extend to other cell types (although likely the pathway would be recruited in response to stimuli other than elevation of network activity). We now discuss this exciting possibility in the discussion section of the revised manuscript.

4. Changes in PV cells activity might be due the result of seizures rather than the other way around making the conclusions that we can drive from Figure 4 ambiguous.

We thank the reviewer for bringing up this very important point and apologize that we had not clarified this in the original manuscript. We carefully monitored 71 mutant mice over multiple weeks. This led to the conclusion that spontaneous seizures occur only in a fraction of mice (12 out of 71 conditional knockout mice as reported in Figure 5h). Animals where a seizure was observed were excluded from the analysis of parvalbumin staining intensity, WFA intensity, innervation, and electrophysiological properties (presented in Figures 4 and 5).

Notably, we monitored 8 conditional knock-out mice in long-term EEG recordings. All seizures in these 8 animals observed over the 14 days of continuous recording occurred during cage changes. Thus, we are confident that we could indeed identify and (thus) exclude all (if not at least the vast majority of) mice that experienced seizures. We now explicitly stated this information in the revised manuscript (Methods section and Figure legend).

Literature cited:

- Donato F, Rompani SB, Caroni P (2013) Parvalbumin-expressing basket-cell network plasticity induced by experience regulates adult learning. *Nature* 504: 272-276
- Ramsaran AI, Wang Y, Golbabaie A, Aleshin S, de Snoo ML, Yeung BA, Rashid AJ, Awasthi A, Lau J, Tran LM *et al* (2023) A shift in the mechanisms controlling hippocampal engram formation during brain maturation. *Science (New York, NY)* 380: 543-551
- Sorensen AT, Cooper YA, Baratta MV, Weng FJ, Zhang Y, Ramamoorthi K, Fropf R, LaVerriere E, Xue J, Young A *et al* (2016) A robust activity marking system for exploring active neuronal ensembles. *eLife* 5
- Yap EL, Pettit NL, Davis CP, Nagy MA, Harmin DA, Golden E, Dagliyan O, Lin C, Rudolph S, Sharma N *et al* (2021) Bidirectional perisomatic inhibitory plasticity of a Fos neuronal network. *Nature* 590: 115-121
- Zhang Y, Rozsa M, Liang Y, Bushey D, Wei Z, Zheng J, Reep D, Broussard GJ, Tsang A, Tsegaye G *et al* (2023) Fast and sensitive GCaMP calcium indicators for imaging neural populations. *Nature* 615: 884-891

Reviewer Reports on the First Revision:

Referees' comments:

Referee #1 (Remarks to the Author):

The authors have put in considerable effort to further improve their exciting study. The extra experiments to demonstrate cell type-specificity of BMP2-SMAD signaling and lack of effect of Smad1 ablation in young mice are strong additions to the manuscript. The authors have convincingly addressed my concerns and the manuscript is suitable for publication in my view. Below a few remaining minor comments/suggestions that do not require re-review:

Minor point 3: Fig. 1c: Id3 fold change is significant compared to mCherry+CNO, not to hMD4+saline; Smad6 fold change is significant compared to both controls. In the text (page 2 line 67) Id1 and Id3 are listed. This should be corrected to Id1 and Smad6; Id3 could be mentioned too but would need some brief mention of the finding that significance is relative to one of the two control conditions.

Minor point 4: The authors indicate that an error was present in the p-values depicted in Fig. 2f and that they have rechecked the calculation. In the revised manuscript, the statistical description is missing in the figure legend of Fig. 2f. Please include the statistical descriptions for the three graphs in the legends.

Discussion page 7 line 215: 'Moreover, an array of type I and type II BMP receptors are detected across neocortical cell populations. Thus, BMP-SMAD1 signaling might control additional aspects of neuronal cell-cell communication.' This is an intriguing implication of the authors' study (as they point out in response to reviewer 3 point 3) that likely appeals to a non-specialized audience. Can the statement about the expression of an array of BMP receptors across cell types be substantiated with a reference?

Typo Fig. 2b: 'necortex'

Referee #3 (Remarks to the Author):

The authors show that BMP2 targets parvalbumin expressing (PV) interneurons, but the specificity of this mechanism for this particular interneuron subtype is unclear. This action occurs through the transcription factor SMAD1, which regulates various glutamatergic synapse proteins. Disrupting BMP2-SMAD1 signaling specifically in PV interneurons results in several consequences, including a reduction in PV cell glutamatergic innervation, underdeveloped peri-neuronal nets, and diminished excitability.

A limitation lies in the fact that manipulations are done in juvenile and not adult (P90 onwards) animals, leaving uncertainties about whether observed effects are solely due to the adult function of BMP2-SMAD1 or if there is a developmental effect. The study shows that disruption of BMP2-SMAD1 signaling in PV interneurons leads to adverse effects, such as a loss of PV cell glutamatergic innervation and underdeveloped peri-neuronal nets, contributing to decreased excitability. However, the potential developmental impact on these outcomes is not adequately addressed.

The link between disrupted cortical excitation-inhibition balance and spontaneous epileptic seizures raises concerns about the study's ability to accurately interpret the observed effects. Many, if not all the results shown could be a consequence of seizure activity. The authors made an attempt to address the issue, however without 24/7h EEG monitoring of the mice it is not possible to rule out

that mice seize at some point. Of notice, some seizures, such as absence seizures, are very hard to detect without EEG monitoring.

The authors, while making a good effort to address reviewer comments, fail to enhance the study's impact substantially. Consequently, the recommendation for publication in a specialized journal appears more fitting, considering the limited advancement in the study's overall significance

Referee #4 (Remarks to the Author):

The manuscript by Okur et al describes a role for BMP signaling in ensuring the proper excitatory-inhibitory balance in a mini-circuit of the adult mouse neocortex. The high potency of BMPs, their very low levels and broad expression patterns, their ability to signal both as homo- and heterodimers, combined with the sheer number and diversity of receiving cells complicate any analysis of intercellular BMP signaling, especially in the CNS. The best one can do is to isolate in space and in time the relevant population(s) of target cells and probe for possible source(s) of ligands. The authors used a battery of elegant and powerful genetic tools to do just that: Upon learning that BMP2 was elevated in glutamatergic cells in response to stimulation, they focused on a population of PV interneurons that seem to respond better than their neighbors to BMP signaling. Within this microcircuit, the authors found that BMP2 secreted from glutamatergic pyramidal cells likely signals to PV interneurons to modulate their function. They report that loss of BMP signaling in PV interneurons down-regulates (1) the number of postsynaptic glutamatergic synapses and (2) the peri-neuronal nets around these neurons and their excitability. These data are consistent with the observed transcriptional changes as determined by ChIP-seq in neocortex and neocortical cultures.

The previous reviewers raised questions and concerns especially regarding the specificity and directionality of the BMP signaling. The revised manuscript includes extensive and thorough responses to these concerns. I was impressed with the extent and the quality of the revisions. This manuscript is a genetics tour de force which describes how BMP signaling influences the engagement of PV interneurons and their function in maintaining an appropriate excitatory-inhibitory balance.

Major points:

- 1) In my view the conclusion that BMP signaling positively modulates glutamatergic input in PV interneurons is well supported. The changes in excitability are only accounted for via Brevican and PNN modifications. What about ion channels and their regulations by BMP signaling? The supplementary tables are rich in a variety of channels that show differential expression in various experimental conditions. The authors should mine their data and further explain the changes in excitability they observe in response to BMP signaling. Limiting this argument to Brevican/PNN fluctuations is a weak point of the manuscript and seems like a lost opportunity.
- 2) A second major concern is one regarding the circuitry: The authors state that BMP2 functions as a relay mechanism, but how does stimulation trigger elevated BMP expression in the first place? Solving this problem may be beyond the scope of this manuscript; nonetheless, the authors should acknowledge the limitations of the current study and discuss possible mechanisms for activity-dependent regulation of BMP2.
- 3) Is parvalbumin a direct target of BMP signaling? The data in Fig. 5a-b and Extended Fig. 8 suggest this may be the case. Please explain.
- 4) The story needs a model/diagram capturing the take-home message, a summary of the

regulation and function of this microcircuit. Such model will greatly facilitate the understanding of this phenomenon and will enhance reader's ability to appreciate not only the biological significance but also the ingenuity of the experimental setup.

Author Rebuttals to First Revision:

Referee #1:

Minor point 3: Fig. 1c: Id3 fold change is significant compared to mCherry+CNO, not to hMD4+saline; Smad6 fold change is significant compared to both controls. In the text (page 2 line 67) Id1 and Id3 are listed. This should be corrected to Id1 and Smad6; Id3 could be mentioned too but would need some brief mention of the finding that significance is relative to one of the two control conditions.

We modified the text accordingly.

Minor point 4: The authors indicate that an error was present in the p-values depicted in Fig. 2f and that they have rechecked the calculation. In the revised manuscript, the statistical description is missing in the figure legend of Fig. 2f. Please include the statistical descriptions for the three graphs in the legends.

We have added this information.

Discussion page 7 line 215: 'Moreover, an array of type I and type II BMP receptors are detected across neocortical cell populations. Thus, BMP-SMAD1 signaling might control additional aspects of neuronal cell-cell communication.' This is an intriguing implication of the authors' study (as they point out in response to reviewer 3 point 3) that likely appeals to a non-specialized audience. Can the statement about the expression of an array of BMP receptors across cell types be substantiated with a reference?

We have added a reference.

Typo Fig. 2b: 'necortex'

Corrected.

Referee #3

A limitation lies in the fact that manipulations are done in juvenile and not adult (P90 onwards) animals, leaving uncertainties about whether observed effects are solely due to the adult function of BMP2-SMAD1 or if there is a developmental effect. The study shows that disruption of BMP2-SMAD1 signaling in PV interneurons leads to adverse effects, such as a loss of PV cell glutamatergic innervation and underdeveloped peri-neuronal nets, contributing to decreased excitability. However, the potential developmental impact on these outcomes is not adequately addressed.

In the original manuscript, we mostly used mice 6-8 weeks of age with readouts collected at 9-11 weeks. In the revised manuscript, several experiments use conditional genetic manipulation at 5-6 weeks (Fig.3 and Extended data Figure 8) and subsequent read-outs at 8-10 weeks of age. The exact definition of juvenile versus adult stages in mice is a topic of discussion. However, one guideline (based on sexual maturity, behavioral and functional criteria) has been to consider mice up to 1 month of age as juvenile, up to 6 months as young adults, and animals aged more than 14 months as 'old'. We chose the timepoint of 5-6 weeks as it is past the age where mice are considered juvenile and – most importantly – after the critical period of development of cortical circuits (PMID:28779074 and 35463203).

The link between disrupted cortical excitation-inhibition balance and spontaneous epileptic seizures raises concerns about the study's ability to accurately interpret the observed effects. Many, if not all the results shown could be a consequence of seizure activity. The authors made an attempt to address the issue, however without 24/7h EEG monitoring of the mice it is not possible to rule out that mice seize at some point. Of notice, some seizures, such as absence seizures, are very hard to detect without EEG monitoring.

This might be a mis-understanding. We did perform and report "24/7" long-term EEG recordings in mice, monitoring and analyzing behavior and EEG activity continuously for multiple weeks. Thus – as stated in the response to reviewers from the revised manuscript we are confident that we could indeed detect all seizures in the mice. We now stated the continuous EEG recording at several additional places of the manuscript to avoid such a mis-understanding by other readers.

Referee #4

The manuscript by Okur et al describes a role for BMP signaling in ensuring the proper excitatory-inhibitory balance in a mini-circuit of the adult mouse neocortex. The high potency of BMPs, their very low levels and broad expression patterns, their ability to signal both as homo- and heterodimers, combined with the sheer number and diversity of receiving cells complicate any analysis of intercellular BMP signaling, especially in the CNS. The best one can do is to isolate in space and in time the relevant population(s) of target cells and probe for possible source(s) of ligands. The authors used a battery of elegant and powerful genetic tools to do just that: Upon learning that BMP2 was elevated in glutamatergic cells in response to stimulation, they focused on a population of PV interneurons that seem

to respond better than their neighbors to BMP signaling. Within this microcircuit, the authors found that BMP2 secreted from glutamatergic pyramidal cells likely signals to PV interneurons to modulate their function. They report that loss of BMP signaling in PV interneurons down-regulates (1) the number of postsynaptic glutamatergic synapses and (2) the peri-neuronal nets around these neurons and their excitability. These data are consistent with the observed transcriptional changes as determined by ChIP-seq in neocortex and neocortical cultures.

The previous reviewers raised questions and concerns especially regarding the specificity and directionality of the BMP signaling. The revised manuscript includes extensive and thorough responses to these concerns. I was impressed with the extent and the quality of the revisions. This manuscript is a genetics tour de force which describes how BMP signaling influences the engagement of PV interneurons and their function in maintaining an appropriate excitatory-inhibitory balance.

Major points:

1) In my view the conclusion that BMP signaling positively modulates glutamatergic input in PV interneurons is well supported. The changes in excitability are only accounted for via Brevican and PNN modifications. What about ion channels and their regulations by BMP signaling? The supplementary tables are rich in a variety of channels that show differential expression in various experimental conditions. The authors should mine their data and further explain the changes in excitability they observe in response to BMP signaling. Limiting this argument to Brevican/PNN fluctuations is a weak point of the manuscript and seems like a lost opportunity.

We fully agree with the reviewer that it is unlikely, that the PV interneuron phenotype can be reduced to a single SMAD1 target gene. Given that we do not provide experimental data on other candidate targets of the SMAD1 transcriptional program we were hesitant to highlight them. We now added a statement in the discussion to address this point.

2) A second major concern is one regarding the circuitry: The authors state that BMP2 functions as a relay mechanism, but how does stimulation trigger elevated BMP expression in the first place? Solving this problem may be beyond the scope of this manuscript; nonetheless, the authors should acknowledge the limitations of the current study and discuss possible mechanisms for activity-dependent regulation of BMP2.

We hypothesize, that upon stimulation, BMP2 is acutely released from dense core vesicles and that the transcriptional up-regulation observed in our experiments represents a re-filling of the vesicular BMP2 pool. We had generated BMP2 knock-in mice carrying an HA epitope tag to investigate this (Extended data Figure 1). However, with our experimental protocols we failed to reliably detect endogenous tagged BMP2 with immunohistochemical methods. We now added discussion on this point in the revised manuscript.

3) Is parvalbumin a direct target of BMP signaling? The data in Fig. 5a-b and Extended Fig. 8 suggest this may be the case. Please explain.

We did not identify SMAD1 binding to parvalbumin promoter elements in our ChIP studies. However, there is extensive literature on the regulation of parvalbumin protein levels by neuronal activity in PV interneurons. Thus, we hypothesize that the alterations in parvalbumin protein levels are secondary to the synaptic and excitability changes. We added a brief comment in the text.

4) The story needs a model/diagram capturing the take-home message, a summary of the regulation and function of this microcircuit. Such model will greatly facilitate the understanding of this phenomenon and will enhance reader's ability to appreciate not only the biological significance but also the ingenuity of the experimental setup.

Thank you for the suggestion – we now added a small cartoon in Figure 5.